# TOWARDS CONSISTENT CROSS-MODAL ALIGNMENT IN CONTINUAL LEARNING FOR VISION-LANGUAGE MODELS

## ABSTRACT

Vision-language models (VLMs) such as CLIP face significant challenges in continual learning (CL), where they must retain both pre-trained and incremental knowledge. Existing methods often rely on reference datasets or domain discriminators, leading to high overhead or limited generalization. Moreover, the semantic gap between modalities hinders effective alignment. While prototypes can partially mitigate this issue, they introduce new challenges: 1) inconsistent prototype fidelity across classes can impede modality fusion and fine-grained alignment, and 2) prototype separability degrades as tasks accumulate in CL. To tackle these, we propose a residual prototype coupled with uncertainty-aware fusion to achieve consistent CLIP alignment. Class-wise prototypes derived from the backbone capture task-specific distributions, supporting both knowledge retention and generalization. Residual prototypes then refine these class representations, mitigating fidelity inconsistency and preserving cross-task separability. In parallel, Bayesian uncertainty-aware estimation and fusion draws on the complementarity between visual prototypes and textual descriptions to dynamically balance multiple objectives, effectively promoting more robust modality fusion and unbiased semantic alignment. Extensive experiments across challenging CL scenarios demonstrate that our method outperforms state-of-the-art approaches, including strong rehearsal-based baselines, across key metrics.

## 1 INTRODUCTION

Existing AI systems are typically built on the assumption of static data distributions and optimize model parameters by minimizing a predefined loss function. However, this assumption breaks down in real-world scenarios where data continuously evolves. Continual learning (CL) has emerged as a prominent solution, requiring models to adapt dynamically to sequential data stream while maintaining performance comparable to joint training on all tasks. Yet, this process often leads to catastrophic forgetting, where knowledge learned from previous is overwritten or distorted (Wang et al., 2024b). To mitigate this, numerous CL methods have been proposed, focusing on stabilizing parameter updates (Kirkpatrick et al., 2017; Li & Hoiem, 2017; Qiao et al., 2023). Recently, with increasing interest in pre-trained models, enabling CL in vision-language models (VLMs) such as CLIP (Jia et al., 2022) has posed new challenges (Yu et al., 2024a). On one hand, achieving effective cross-modal alignment is inherently difficult (Liang et al., 2022; Wang et al., 2024a; Zhu et al., 2024; Zhou et al., 2025b); on the other hand, VLMs must incrementally acquire and retain domain-specific knowledge while simultaneously preserving the general knowledge obtained during pre-training (Zheng et al., 2023; Yu et al., 2024b; Xu et al., 2024).

In CL, CLIP often acts as a source of prior knowledge, offering strong pre-trained representations that are biased toward the original training domains, which limits incremental adaptation and efficient knowledge transfer. ZSCL (Zheng et al., 2023) leverage extra reference datasets and a dual-CLIP setup – one trainable and one frozen – to transfer both pre-trained and incremental knowledge through distillation or related mechanisms (Li & Hoiem, 2017; Wortsman et al., 2022), as illustrated in Figure 1a. Obviously, such methods incur substantial computational and storage overhead. MoE-Adapter (Yu et al., 2024b) and RAIL (Xu et al., 2024) represent two typical adapter-based methods (Figure 1b). MoE-Adapter employs an adapter-based mixture-of-experts architecture to isolate

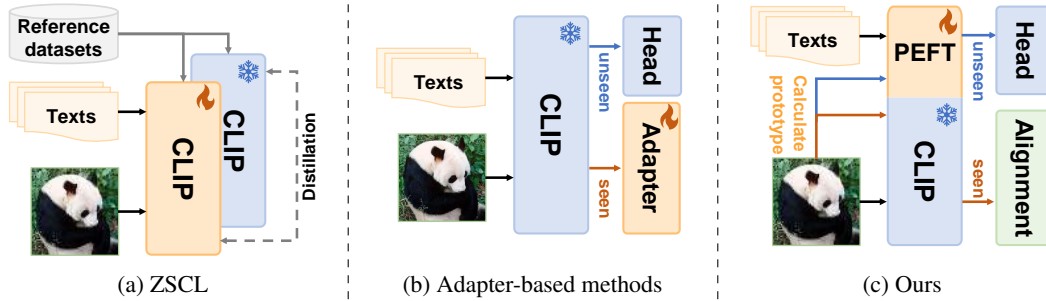

Figure 1: Comparison of different CL methods for CLIP. (a) ZSCL incur high overhead by fine-tuning the whole CLIP with reference data. (b) Adapter-based methods fail to exploit incremental knowledge and are limited by the modality gap. (c) Our method employs excel in both domain generalization and cross-modal alignment.

parameter across domains and further trains a domain discriminator using reference datasets to retain zero-shot performance. Nonetheless, it still relies on external data and fails to exploit incrementally acquired knowledge to improve unseen domain generalization. RAIL adopts a training-free framework based on static ridge regression, inherently limiting knowledge transfer from previously seen tasks (domains). Moreover, it heavily depends on the quality of the representation space. As a result, the computational burden escalates as tasks accumulate, and performance tends to degrade when handling large-scale datasets.

Beyond these, the semantic gap between modalities hinders effective alignment, constraining the performance upper bound. In practice, visual prototypes abstract complex, low-level visual signals into stable fine-grained representations, thereby partially bridging this gap. Methods like Tip-Adapter (Zhang et al., 2021), PROOF (Zhou et al., 2025b), and LADA (Luo et al., 2025) either statically bias toward or simply average certain semantic information. However, visual and textual semantics often provide complementary representations, making it difficult to dynamically estimate their reliability in realistic CL. In addition, prototype fidelity across classes is sensitive to data scale and quality as well as CLIP biased toward the pre-trained domains. Prototype separability, meanwhile, degrades as tasks accumulate in CL, leading to prototype interference (Li et al., 2024). Together, these factors impede modality fusion and fine-grained alignment, exacerbating the difficulty of estimating the uncertainty of different semantics.

In this work, we introduce a residual prototype coupled with uncertainty-aware fusion to enable continual learning of CLIP without rehearsal, as illustrated in Figure 1c. Class-wise prototypes and task-specific prompts are used to model task distributions and capture task-wise semantics, while prototypes from seen domains are further leveraged to adaptively integrate incremental prompts, facilitating generalization to unseen domains. To ensure consistent fidelity and discriminability of prototypes, residual prototypes independently refine class representations, which preserves intra-modality consistency and supports more reliable integration of visual and textual semantics through attention. Building on this, we introduce Bayesian uncertainty-aware estimation and fusion to dynamically balance the reliability of different semantic sources. Unlike static fusion, this enables adaptive fusion of complementary semantics, thereby mitigating biased modality contributions and ensuring more consistent cross-modal alignment in CL. The contributions are summarized as follows:

- We propose a rehearsal-free CL method for CLIP that achieves consistent cross-modal alignment while supporting better generalization to unseen domains.

- We leverage prototypes and task-specific prompts to dynamically aggregate knowledge from seen domains, enhancing knowledge retention and generalization to unseen domains.

- We refine class representations via residual prototypes and leverage a semantic uncertainty perspective through Bayesian estimation and fusion to synergistically balance multiple semantics, promoting robust modality fusion and unbiased cross-modal alignment.

- We conduct extensive experiments demonstrating that our method consistently achieves state-of-the-art results across challenging continual learning settings.

## 2 RELATED WORK

**Continual Learning.** Previous CL approaches have focused mainly on addressing Task-Incremental Learning (TIL) (Ge et al., 2023), Domain-Incremental Learning (DIL) (Wang et al., 2022a; 2024c), and Class-Incremental Learning (CIL) (Li & Hoiem, 2017; Wang et al., 2023; Zhou et al., 2025b). Among them, CIL is a more challenging setting where task identity is not provided during the inference phase, and the number of classes increases as tasks arrive. Replay-based methods (Rebuffi et al., 2017; Van de Ven et al., 2020) store or regenerate a portion of historical data to recover past distributions. Regularization-based methods (Kirkpatrick et al., 2017; Zheng et al., 2023) constrain important parameters to reduce forgetting. Distillation-based methods (Li & Hoiem, 2017; Ding et al., 2022) distill knowledge from previous models to facilitate knowledge transfer. Architecture-based methods (Yu et al., 2024b; Tang et al., 2024) dynamically expand network to isolate parameters. Recently, Zheng et al. (2023) and Xu et al. (2024) proposed two emerging CL settings: Multi-Domain Task-Incremental Learning (MTIL) and Cross-Domain Task-Agnostic Incremental Learning (X-TAIL). Both aim to preserve the pre-trained and incremental knowledge when continually adapting to downstream tasks. Similarly to TIL, MTIL relies on task identity to construct the test label space, while X-TAIL includes both seen and novel classes, thereby better reflecting real-world conditions.

**Parameter-Efficient Fine-Tuning.** PEFT methods were initially proposed in large language models (LLMs) to enable rapid adaptation to downstream tasks with reduced computational overhead (Li & Liang, 2021; Lester et al., 2021; Liu et al., 2021; Hu et al., 2022), later extending to the visual and multi-modal domains (Zhou et al., 2022b; Khattak et al., 2023; Gao et al., 2024). Recently, prompt-based methods have garnered increasing attention as interest grows in CL with pre-trained models. L2P (Wang et al., 2022b) and CODA-Prompt (Smith et al., 2023) introduces a prompt pool that selects or computes task-relevant prompts conditioned on image inputs. AttriCLIP (Wang et al., 2023) and DIKI (Tang et al., 2024) further extend this to CLIP (Radford et al., 2021a). Essentially, Prompt tuning preserves the backbone architecture while injecting a small number of learnable parameters to steer internal representations toward task-specific subspaces. In VLMs, however, its effectiveness hinges on the semantic richness of textual descriptions (Pratt et al., 2023). When such semantics are insufficient, cross-modal alignment can be adversely affected.

**Prototypes in Continual Learning.** In human cognition, we typically abstract and categorize information around conceptual centers rather than memorizing all instances. Prototypes reflect this principle well. Initially, Snell et al. (2017) leveraged prototypes to tackle few- and zero-shot classification. In CL, well-formed prototypes are promising as they can capture class-wise semantics within a shared representation space. iCaRL (Rebuffi et al., 2017) constructs prototypes from a few representative samples stored in memory, and replaces the classifier with a nearest-class-mean strategy to alleviate forgetting caused by parameter coupling. PROOF (Zhou et al., 2025b) further introduces expandable projection to incorporate new concepts while preserving old knowledge with rehearsal, and leverages a fusion module that exploits cross-modal information. CPP (Li et al., 2024) employ prototypes as anchors in the latent task space to prevent prototype interference across tasks, and LADA (Luo et al., 2025) further presents a label-specific CLIP adapter that does not require parameter selection.

## 3 METHODOLOGY

### 3.1 PRELIMINARIES

**Problem Setting.** In a typical CL setting, the model sequentially adapts a series of tasks. For each task $t$, there is an associated dataset $\mathcal{D}^t = \{(x_i^t, y_i^t)\}_{i=1}^{B^t}$, where $x_i^t \in \mathbb{R}^m$ denotes an image sample, $y_i^t$ represents its corresponding class label and $B^t$ indicates the number of samples in task $t$. The category set for task $t$ is defined as the set of category names corresponding to $|\mathcal{C}^t|$ distinct labels, i.e., $\mathcal{C}^t = \{c_i^t\}_{i=1}^{|\mathcal{C}^t|}$. The complete category set across all tasks is $\mathcal{C} = \cup_{i=1}^{N_t}\mathcal{C}^i = \mathcal{C}^{seen} \cup \mathcal{C}^{unseen}$ with $N_t$ total tasks. While MTIL defines the test label space as the domain-specific category subset $\mathcal{C}^t$, X-TAIL evaluates the model over the full category set $\mathcal{C}$ without any domain identity hints.

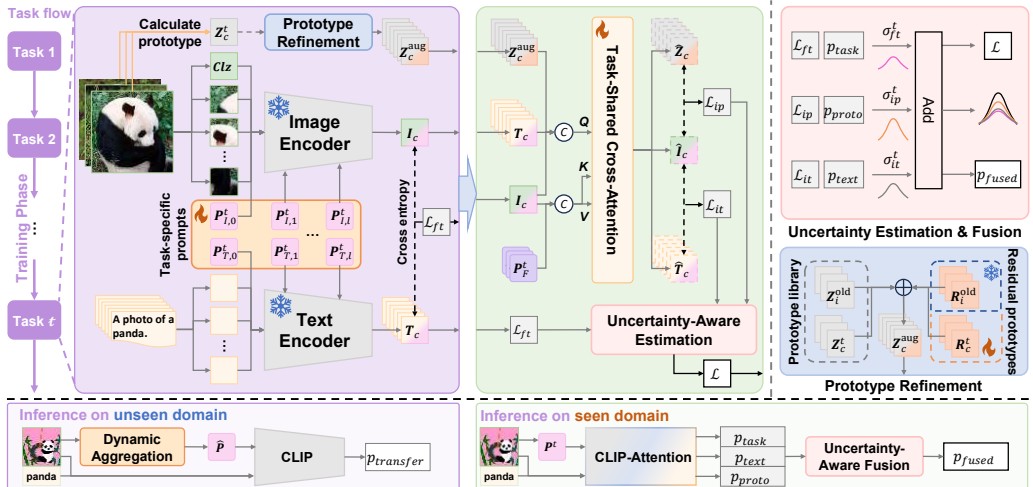

Figure 2: Overview of the proposed method. (a) **Training Phase**: For each task $t$, class prototypes are stored in a prototype library. Task-specific prompts generate image and text features (orange), which are concatenated with the refined prototypes (blue) and then fed into the fusion module. Modality-wise uncertainty-aware estimation is learned as in Eq. 11, 12 (pink). (b) **Inference Phase**: For unseen domains, dynamic aggregation (orange) integrate pre-trained and incremental knowledge via prototypes, as described in Eq. 5, 6. For seen domains, modality-wise uncertainty-aware fusion incorporates the reliable semantic sources, as defined in Eq. 13(pink).

**CLIP Model.** CLIP employs a dual-encoder architecture consisting of an image encoder $f_\theta(\cdot)$ and a text encoder $g_\phi(\cdot)$, which project visual and textual modalities into a shared embedding space: $\mathbb{R}^{D_I} \to \mathbb{R}^d, \mathbb{R}^{D_T} \to \mathbb{R}^d$. Specifically, given an input image $x$, the image encoder extract the visual feature $\boldsymbol{I} = f_\theta(x)$. On the textual side, each category in $\mathcal{C}$ is inserted into one or more predefined templates (e.g., *a photo of a [CLASS NAME].*) to form a set of textual inputs $\mathcal{W} = \{w_i\}_{i=1}^{|\mathcal{C}|}$. The posterior probability of zero-shot is then computed as:

$$p_{\text{zs}}(y_i \mid x) = \frac{\exp\left(\cos\left(\boldsymbol{I}, g_\phi(w_i)\right)/\tau\right)}{\sum_j^{|\mathcal{C}|} \exp\left(\cos\left(\boldsymbol{I}, g_\phi(w_j)\right)/\tau\right)}, \tag{1}$$

where $\cos(\cdot, \cdot)$ denotes the cosine similarity, and $\tau$ is a temperature scaling parameter.

**Prompt Learning.** Prompt tuning (Khattak et al., 2023) can adapt to diverse downstream tasks, thereby effectively capturing task-wise semantics. Thus, we introduce task-specific learnable prompts $\boldsymbol{P}^t = \{(\boldsymbol{P}_{I,i}^t, \boldsymbol{P}_{T,i}^t)\}_{i=0}^{L-1}$, where $\boldsymbol{P}_{I,i}^t \in \mathbb{R}^{L_p \times d_I}$ and $\boldsymbol{P}_{T,i}^t \in \mathbb{R}^{L_p \times d_T}$ are inserted into the input embeddings of the $i$-th transformer block in the image and text encoders, respectively. $L_p$ is the prompt length, $L$ is the total number of transformer layers, $d_I$ and $d_T$ are the dimensions of the image and text token. Let $\mathcal{B}_{I,l}$ and $\mathcal{B}_{T,l}$ denote the $l$-th transformer block of the image and text encoders, respectively. For the first $L$ layers ($l = 1, ..., L$), $\boldsymbol{P}^t$ are injected before each layer as follows:

$$\left[\boldsymbol{c}_l, \_, \boldsymbol{V}_l\right] = \mathcal{B}_{I,l}\left(\left[\boldsymbol{c}_{l-1}, \boldsymbol{P}_{I,l-1}^t, \boldsymbol{V}_{l-1}\right]\right), \quad \left[\_, \boldsymbol{S}_l\right] = \mathcal{B}_{T,l}\left(\left[\boldsymbol{P}_{T,l-1}^t, \boldsymbol{S}_{l-1}\right]\right). \tag{2}$$

For subsequent layers, the prompts are not updated. Here, $\boldsymbol{c}_l \in \mathbb{R}^{1 \times D_I}$ is the class token, $\boldsymbol{V} \in \mathbb{R}^{L_I \times d_I}$ is the fixed image tokens, and $\boldsymbol{S} \in \mathbb{R}^{L_T \times d_T}$ is the fixed text tokens, with $L_I$ and $L_T$ denoting their respective token lengths. As shown in Figure 2 (orange), image and text features are then computed as $\boldsymbol{I} = f_\theta(x; \boldsymbol{P}_I^t)$ and $\boldsymbol{T} = g_\phi(\mathcal{W}^t; \boldsymbol{P}_T^t)$, where $\mathcal{W}^t = \{w_i\}_{i=1}^{M^t}$, and $M^t = \sum_i^t |\mathcal{C}^i|$ is the total number of seen classes. The probability follows Eq. 1 and is computed under task-wise semantics as:

$$p_{\text{pt}}(y_i \mid x) = \frac{\exp\left(\cos\left(\boldsymbol{I}, \boldsymbol{T}_i\right)/\tau\right)}{\sum_j^{M^t} \exp\left(\cos\left(\boldsymbol{I}, \boldsymbol{T}_j\right)/\tau\right)} \tag{3}$$

### 3.2 FROM TASK-SPECIFIC TO CROSS-TASK PROMPTS: RETHINKING CLIP SEMANTIC ALIGNMENT

Leveraging the strong representation of pre-trained models, approaches like L2P (Wang et al., 2022b) and DIKI (Tang et al., 2024) exploit it to identify a task-specific subspace. In a similar spirit, we compute class-wise visual prototypes before training to preserve task distribution over time, formulated as $\boldsymbol{Z}_c^t = \frac{1}{|\mathcal{D}_c^t|}\sum_{x\in\mathcal{D}_c^t} f_\theta(x)$, where $\mathcal{D}_c^t$ denotes all samples belonging to class $c$ in task $t$.

These prototypes are then stored in a prototype library $\mathcal{P} = \{\boldsymbol{Z}^j \in \mathbb{R}^{|\mathcal{C}^j|\times d}\}_j^{N_t}$. This ensures the representation of each task remains accurate and discriminative regardless of the data scale.

During inference, we first identify the input's domain with CLIP. If the domain is seen, we extract the image feature $\boldsymbol{q}$ and identify the most likely task by measuring similarity with historical prototypes:

$$\hat{t} = \underset{t\in\{1,\dots,N_t\},\, c\in\mathcal{C}^t}{\arg\max} \left\{\cos\left(\boldsymbol{q}, \boldsymbol{Z}_c^t\right)\right\}, \tag{4}$$

**Theorem 1** (Weighted aggregation outperforms single prompt in expectation). *Assuming that model's output probability $p\left(x; \boldsymbol{P}\right)$ w.r.t. $\boldsymbol{P}$ is an approximately concave function, Jensen's inequality implies that weighted prompt aggregation yields a higher expected probability than any individual one:*

$$p\left(x; \hat{\boldsymbol{P}}\right) \geq \sum_i \varphi^i \cdot p\left(x; \boldsymbol{P}^i\right) \geq \varphi^j \cdot p\left(x; \boldsymbol{P}^j\right), \quad j = \arg\max_{j\in\{1,\dots,t\}}\left\{\varphi^j\right\}.$$

If the domain is unseen, selecting an optimal prompt $\hat{\boldsymbol{P}}$ becomes nontrivial since prompts $\boldsymbol{P}$ are not explicitly optimized for unseen domains. Empirically, prompt tuning activates and modulates internal attention patterns, enabling the model to capture more domain-aligned inductive biases without significantly degrading performance on novel classes (Zhou et al., 2022a;b). To leverage knowledge from both seen tasks and pre-trained domains, we propose to aggregate all previously learned prompts, each weighted by a similarity score $\varphi^i$, to form a composite representation (Theorem 1), as shown in Figure 2 (orange).

$$\varphi^i = \frac{\sum_c^{|\mathcal{C}^i|}\cos(\boldsymbol{q}, \boldsymbol{Z}_c^t)}{\sum_i^t \sum_c^{|\mathcal{C}^i|}\cos(\boldsymbol{q}, \boldsymbol{Z}_c^t)}, \quad \hat{\boldsymbol{P}} = \sum_i^t \varphi^i \cdot \boldsymbol{P}^i. \tag{5}$$

Meanwhile, we introduce a hyperparameter $\alpha \in (0,1)$ to balance the generalization ability of the original model ($\alpha \to 1$) and the domain adaptation of the fine-tuned model ($\alpha \to 0$):

$$p_{\text{transfer}}(x) = \alpha \cdot p_{\text{zs}}(x) + (1-\alpha) \cdot p_{\text{pt}}(x; \hat{\boldsymbol{P}}). \tag{6}$$

**Visual prototypes vs. textual features: which contains richer semantic information?** To investigate this, we adopt a train-free baseline, nearest-class-mean (NCM) (Rebuffi et al., 2017), as a naive classifier:

$$\hat{y} = \underset{y\in\{1,\dots,|\mathcal{C}|\}}{\arg\max}\left\{\cos\left(\boldsymbol{q}, \boldsymbol{Z}_y\right)\right\}. \tag{7}$$

We compare its classification accuracy with CLIP zero-shot ability in the X-TAIL setting. As shown in Figure 3, the former outperforms the latter on most domains, even in few-shot scenarios, particularly when the original class descriptions are insufficient or the domains are highly fine-grained or professional, such as Aircraft (Maji et al., 2013) and EuroSAT (Helber et al., 2019). Theoretically, text features have the potential to encode rich semantic knowledge via LLMs, but this advantage often relies on carefully designed prompts or external knowledge injection (Pratt et al., 2023; Yang et al., 2023; Zhou et al., 2025a), providing semantic information more aligned with visual tasks.

Figure 3: Cross-domain accuracy of CLIP zero-shot vs. NCM classifiers under different few-shot settings.

### 3.3 RESIDUAL PROTOTYPES FOR CROSS-MODAL FUSION

In practice, visual prototypes and textual descriptions provide complementary and diverse semantics: the former encapsulates fine-grained visual characteristics grounded in real data, while the latter

conveys abstract, high-level knowledge from language, and their relative reliability may vary across scenarios. Intuitively, their dynamic integration can offer a potential optimization path for cross-modal alignment. Although PROOF (Zhou et al., 2025b) adopts a cross-modal fusion that enables mutual semantic exchange between modalities, such methods typically assume consistent prototype fidelity, which rarely holds in CL: class-wise visual prototypes vary with data scale, quality and backbone's pre-training bias, and their separability degrades as tasks accumulate, leading to prototype interference. To address this, we introduce class-wise learnable residual prototypes that refine the original class representations on a token-wise basis:

$$\boldsymbol{Z}^{\mathrm{aug}} = \boldsymbol{Z} + \boldsymbol{R}, \tag{8}$$

where $\boldsymbol{R}$ is initialized as $\boldsymbol{0}$ and frozen for past tasks. During training, residual prototypes from previous tasks are kept frozen as shown in Figure 2 (blue). This lightweight design improves intra-modality consistency, thereby eliminating intra-modal bias and supporting more reliable integration of visual and textual semantics through cross-modal fusion. Moreover, residual prototypes implicitly encode task-specific knowledge, enabling us to employ a single task-shared attention module without rehearsal, while still mitigating forgetting in CL.

Now, in task $t$, the inputs consist of image features $\boldsymbol{I} \in \mathbb{R}^{1 \times d}$ and text features $\boldsymbol{T} \in \mathbb{R}^{M^t \times d}$, both containing task-specific semantics, together with refined prototypes $\boldsymbol{Z}^{\mathrm{aug}} \in \mathbb{R}^{M^t \times d}$. These are concatenated to form the query $\boldsymbol{Q}$. For constructing key and value, we introduce cross-task fusion prompts $\boldsymbol{P}_F \in \mathbb{R}^{(t \times L_F) \times d}$ to facilitate interaction between cross-modal features while efficiently integrating accumulated semantic information across tasks, reducing computational overhead. Accordingly, $\boldsymbol{K}$ and $\boldsymbol{V}$ are formed by concatenating $\boldsymbol{I}$ and $\boldsymbol{P}_F$. They are subsequently processed by a task-shared cross-attention module.

$$\boldsymbol{X} = \mathrm{Attention}\left(\boldsymbol{Q}, \boldsymbol{K}, \boldsymbol{V}\right), \tag{9}$$

where $\boldsymbol{X} = [\hat{\boldsymbol{I}}, \hat{\boldsymbol{T}}, \hat{\boldsymbol{Z}}]$ corresponds to the enhanced image features, text features, and prototypes, respectively. Following Eq. 3, we compute probabilities under textual and visual semantic objectives as:

$$p_{\mathrm{it}}\left(y_i \mid x\right) = \frac{\exp\left(\cos\left(\hat{\boldsymbol{I}}, \hat{\boldsymbol{T}}_i\right)/\tau\right)}{\sum_j^{M^t} \exp\left(\cos\left(\hat{\boldsymbol{I}}, \hat{\boldsymbol{T}}_j\right)/\tau\right)}, \quad p_{\mathrm{ip}}\left(y_i \mid x\right) = \frac{\exp\left(\cos\left(\hat{\boldsymbol{I}}, \hat{\boldsymbol{Z}}_i\right)/\tau\right)}{\sum_j^{M^t} \exp\left(\cos\left(\hat{\boldsymbol{I}}, \hat{\boldsymbol{Z}}_j\right)/\tau\right)}. \tag{10}$$

Finally, to futher mitigate forgetting in the task-shared module, we use Elastic Weight Consolidation (EWC) (Kirkpatrick et al., 2017), which constrains updates to parameters deemed important.

### 3.4 BAYESIAN UNCERTAINTY-AWARE ESTIMATION FOR CONSISTENT FUSION

Now, we obtain three semantic objectives – $p_{\mathrm{pt}}$, $p_{\mathrm{it}}$, and $p_{\mathrm{ip}}$ – with corresponding loss functions:

$$\mathcal{L}_1 = -\frac{1}{N}\sum_i^N \log p_{\mathrm{pt}}\left(y_i \mid x_i\right), \; \mathcal{L}_2 = -\frac{1}{N}\sum_i^N \log p_{\mathrm{it}}\left(y_i \mid x_i\right), \; \mathcal{L}_3 = -\frac{1}{N}\sum_i^N \log p_{\mathrm{ip}}\left(y_i \mid x_i\right). \tag{11}$$

Previous methods often assume equal reliability across modalities, but in practice, the reliability of different semantic sources varies. To dynamically balance inter-modal contributions, we leverage the intra-modality consistency provided by residual prototypes and draw inspiration from Bayesian uncertainty theory (Kendall et al., 2018), estimating modality-wise homoscedastic uncertainty. As illustrated in Figure 2 (pink), for each task $t$, we learn an uncertainty coefficient $\boldsymbol{\sigma}^t = \{\sigma_1^t, \sigma_2^t, \sigma_3^t\}$ to adaptively modulate the contribution of each semantic objective to the overall objective $\mathcal{L}_{\mathrm{total}}$. Incorporating EWC with the penalty coefficient $\lambda$, the final loss is defined as $\mathcal{L}$.

$$\mathcal{L}_{\mathrm{total}} = \frac{1}{2} \cdot \sum_k^3 \frac{1}{(\sigma_k^t)^2}\mathcal{L}_k + \log \sigma_k^t, \quad \mathcal{L} = \mathcal{L}_{\mathrm{total}} + \lambda \cdot \mathcal{L}_{\mathrm{EWC}} \tag{12}$$

**Theorem 2** (Uncertainty Estimation). *Assume consistent observation noise between training and testing. Let the true logit $z \sim \mathcal{N}(\mu_0, \sigma_0^2)$, and suppose each of the $K$ semantic objectives independently observes a noisy version $z_k \sim \mathcal{N}(z, \sigma_k^2)$. Under a uniform prior, the MAP estimate of $z$ reduces to:*

$$\hat{z} = \frac{\mu_0/\sigma_0^2 + \sum_{k=1}^K z_k/\sigma_k^2}{1/\sigma_0^2 + \sum_{k=1}^K 1/\sigma_k^2} \approx \frac{\sum_{k=1}^K z_k/\sigma_k^2}{\sum_{k=1}^K 1/\sigma_k^2} \propto \sum_{k=1}^K \frac{z_k}{\sigma_k^2}.$$

Table 1: Comparison of different CL methods on 16-shot X-TAIL for each domain in terms of *Transfer*, *Average*, and *Last* scores (%). The best and second results are highlighted with **bold** and underline styles, respectively. RAIL[*] includes both Primal- and Dual-RAIL, as they adopt the same domain discriminator technique. Dual-RAIL[†] stores and replays all historical data.

| | Method | Aircraft | Caltech101 | DTD | EuroSAT | Flowers | Food | MNIST | Pets | Cars | SUN397 | Average |
|---|---|---|---|---|---|---|---|---|---|---|---|---|
| | Zero-shot | 24.8 | 73.9 | 37.5 | 52.0 | 71.0 | 88.0 | 42.5 | 88.8 | 64.0 | 60.2 | 60.3 |
| **Transfer** | LwF (Li & Hoiem, 2017) | – | 72.3 | 33.0 | 27.2 | 51.5 | 76.9 | 30.6 | 74.4 | 35.3 | 54.6 | 50.6 |
| | iCaRL (Rebuffi et al., 2017) | – | 54.0 | 24.8 | 15.3 | 40.7 | 65.2 | 30.2 | 72.2 | 17.6 | 48.2 | 40.9 |
| | WiSE-FT (Wortsman et al., 2022) | – | 69.6 | 31.7 | 40.0 | 47.8 | 75.2 | 20.4 | 75.0 | 38.8 | 54.5 | 50.3 |
| | ZSCL (Zheng et al., 2023) | – | 75.1 | **38.5** | 44.2 | 66.5 | 87.4 | 27.5 | 87.2 | 58.6 | 62.7 | 60.9 |
| | MoE-Adapter (Yu et al., 2024b) | – | 73.9 | 34.9 | 52.0 | 69.4 | 88.0 | 43.4 | 88.8 | **64.0** | 60.2 | 63.8 |
| | RAIL[*] (Xu et al., 2024) | – | 73.9 | 37.5 | 52.0 | 71.0 | 88.0 | 42.5 | 88.8 | **64.0** | 60.2 | 64.2 |
| | Ours | – | **75.7** | 37.2 | **52.1** | **71.1** | **88.3** | **43.7** | **89.2** | 63.7 | **62.9** | **64.9** |
| **Average** | LwF (Li & Hoiem, 2017) | 29.9 | 84.3 | 50.6 | 47.9 | 69.4 | 75.2 | 57.6 | 75.2 | 41.2 | 56.3 | 58.8 |
| | iCaRL (Rebuffi et al., 2017) | 43.1 | 73.8 | 46.3 | 35.6 | 60.1 | 67.9 | 56.8 | 76.9 | 29.0 | 50.3 | 54.0 |
| | WiSE-FT (Wortsman et al., 2022) | 44.8 | 85.0 | 55.7 | 67.5 | 72.6 | 78.8 | 51.2 | 79.3 | 47.2 | 56.4 | 63.9 |
| | ZSCL (Zheng et al., 2023) | 40.1 | 78.2 | 56.8 | 71.8 | 82.4 | 88.4 | 50.4 | 88.5 | 63.3 | 63.9 | 68.4 |
| | MoE-Adapter (Yu et al., 2024b) | 44.6 | 82.0 | **61.9** | 52.0 | 85.1 | 87.8 | **64.2** | 88.8 | 67.4 | 61.4 | 69.5 |
| | Primal-RAIL (Xu et al., 2024) | 45.3 | 87.8 | 58.1 | 76.2 | 86.0 | 89.0 | 62.4 | 89.9 | 67.4 | 61.5 | 72.4 |
| | Dual-RAIL[†] (Xu et al., 2024) | 46.2 | 87.9 | 59.0 | 76.8 | 86.2 | **89.1** | 62.9 | 89.9 | **67.6** | 61.6 | 72.7 |
| | Ours | **49.6** | **88.0** | 60.0 | **78.7** | **86.7** | 88.4 | 63.9 | **90.5** | **67.6** | **64.0** | **73.7** |
| **Last** | LwF (Li & Hoiem, 2017) | 25.1 | 81.9 | 53.5 | 66.2 | 80.4 | 76.6 | 97.9 | 81.0 | 62.3 | 72.1 | 69.7 |
| | iCaRL (Rebuffi et al., 2017) | 43.4 | 77.3 | 53.1 | 45.4 | 78.8 | 76.5 | 97.5 | 87.1 | 74.2 | 69.2 | 70.3 |
| | WiSE-FT (Wortsman et al., 2022) | 36.4 | 84.5 | 57.4 | 65.7 | 88.0 | 81.8 | **98.3** | 88.5 | 78.9 | 73.1 | 75.3 |
| | ZSCL (Zheng et al., 2023) | 36.0 | 79.7 | 59.7 | 79.3 | 90.9 | 89.5 | 84.9 | 91.1 | 81.8 | **74.7** | 76.8 |
| | MoE-Adapter (Yu et al., 2024b) | 44.4 | 81.4 | 68.6 | 52.0 | 95.1 | 87.7 | 95.2 | 88.9 | 81.1 | 72.8 | 76.7 |
| | Primal-RAIL (Xu et al., 2024) | 44.6 | 94.1 | 68.0 | 86.4 | 95.5 | 90.1 | 91.3 | 92.6 | 80.9 | 73.5 | 81.7 |
| | Dual-RAIL[†] (Xu et al., 2024) | 46.1 | **94.8** | **70.2** | 88.2 | **96.3** | **90.2** | 93.2 | 92.6 | 82.2 | 73.9 | 82.8 |
| | Ours | **49.1** | 92.9 | 70.0 | **90.7** | 97.2 | 88.5 | 93.9 | **93.5** | **83.0** | 74.2 | **83.3** |

Finally, grounded on Theorem 2, we design a modality-wise probabilistic fusion mechanism that incorporates all semantic objectives in a uncertainty-aware manner during inference.

$$\tilde{s}_i = \frac{\cos(\boldsymbol{I}, \boldsymbol{T}_i)}{(\sigma_1^t)^2} + \frac{\cos(\hat{\boldsymbol{I}}, \hat{\boldsymbol{T}}_i)}{(\sigma_2^t)^2} + \frac{\cos(\hat{\boldsymbol{I}}, \hat{\boldsymbol{Z}}_i)}{(\sigma_3^t)^2}, \quad p_{\text{fused}}(y_i \mid x) = \frac{\exp(\tilde{s}_i)}{\sum_j \exp(\tilde{s}_j)}. \quad (13)$$

This facilitates adaptive fusion of complementary semantics, mitigating biased influences from individual modalities and promoting more consistent CLIP alignment in CL.

## 4 EXPERIMENTS

### 4.1 EXPERIMENT SETTING

**Datasets.** Following Xu et al. (2024), we evaluate our method under two challenging continual learning scenarios: MTIL and X-TAIL. In the MTIL setting, 11 datasets are involved in alphabetical order: Aircraft (Maji et al., 2013), Caltech101 (Fei-Fei et al., 2004), CIFAR100 (Krizhevsky et al., 2009), DTD (Cimpoi et al., 2014), EuroSAT (Helber et al., 2019), Flowers (Nilsback & Zisserman, 2008), Food (Bossard et al., 2014), MNIST (Deng, 2012), OxfordPet (Parkhi et al., 2012), Stanford-Cars (Krause et al., 2013), and SUN397 (Xiao et al., 2010), resulting in 1200 classes. For X-TAIL, we follow Xu et al. (2024) and exclude CIFAR100 to reduce domain overlap with other datasets, resulting in 1100 classes. We construct few-shot variants by sampling 5-shot training subsets for MTIL and 16-shot for X-TAIL, while retaining the full test sets.

**Evaluation Metrics.** In line with prior works (Xu et al., 2024), we leverage three metrics, namely *Transfer*, *Average*, and *Last*, to evelute our method. *Transfer* measures the zero-shot transfer performance of the model on unseen domains. *Last* assesses the model's ability to retain knowledge from all previously learned tasks. *Average* represents the cumulative average of *Transfer* and *Last* metrics across all test domains during training process.

Table 2: Comparison of different CL methods on 5-shot MTIL for each domain in terms of *Transfer*, *Average*, and *Last* scores (%). The best and second results are highlighted with **bold** and underline styles, respectively.

| | Method | Aircraft | Caltech101 | CIFAR100 | DTD | EuroSAT | Flowers | Food | MNIST | Pets | Cars | SUN397 | Average |
|---|---|---|---|---|---|---|---|---|---|---|---|---|---|
| | Zero-shot | 24.8 | 93.5 | 68.4 | 42.9 | 54.9 | 71.0 | 88.5 | 59.4 | 89.1 | 64.0 | 61.6 | 65.3 |
| **Transfer** | LwF (Li & Hoiem, 2017) | – | 86.8 | 64.6 | 41.2 | 44.0 | 56.0 | 77.1 | 61.4 | 75.4 | 36.8 | 53.5 | 59.7 |
| | iCaRL (Rebuffi et al., 2017) | – | 66.2 | 45.6 | 46.7 | 27.7 | 45.0 | 66.1 | 46.9 | 78.0 | 23.6 | 49.2 | 48.1 |
| | WiSE-FT (Wortsman et al., 2022) | – | 86.4 | 62.3 | 42.3 | 36.5 | 52.7 | 78.6 | 62.9 | 77.9 | 43.5 | 58.7 | 60.2 |
| | ZSCL (Zheng et al., 2023) | – | 92.4 | 66.7 | **44.5** | 47.3 | 66.8 | 87.0 | 63.3 | 85.7 | 56.2 | 62.2 | 67.2 |
| | MoE-Adapter (Yu et al., 2024b) | – | 93.5 | 68.4 | 41.8 | 47.3 | 68.3 | 88.5 | 60.4 | 89.1 | **64.0** | 61.6 | 68.3 |
| | RAIL (Xu et al., 2024) | – | 93.5 | 68.4 | 42.9 | 54.9 | 71.0 | 88.5 | 59.4 | 89.1 | **64.0** | 61.6 | 69.3 |
| | Ours | – | **94.1** | **69.4** | 43.7 | **55.4** | **71.3** | **89.0** | 57.8 | **89.5** | **64.0** | **64.1** | **69.8** |
| **Average** | LwF (Li & Hoiem, 2017) | 22.2 | 90.2 | 61.3 | 50.4 | 61.0 | 66.9 | 72.8 | 73.8 | 75.0 | 40.9 | 54.7 | 60.8 |
| | iCaRL (Rebuffi et al., 2017) | 25.2 | 84.9 | 51.2 | 46.7 | 50.5 | 58.6 | 67.0 | 64.1 | 80.3 | 30.4 | 50.8 | 55.4 |
| | WiSE-FT (Wortsman et al., 2022) | 32.3 | 92.8 | 61.7 | 56.0 | 63.2 | 69.0 | 78.8 | **75.4** | 80.3 | 47.7 | 59.6 | 65.2 |
| | ZSCL (Zheng et al., 2023) | 27.9 | 93.3 | 73.0 | 57.3 | 67.4 | 79.1 | 87.2 | 74.3 | 86.5 | 58.9 | 63.1 | 69.8 |
| | MoE-Adapter (Yu et al., 2024b) | 32.3 | 95.3 | **74.8** | **61.2** | 42.8 | 83.0 | 88.2 | 60.4 | 89.1 | 64.5 | 62.3 | 68.5 |
| | Primal-RAIL (Xu et al., 2024) | 34.4 | 95.5 | 67.6 | 57.1 | 72.3 | 83.7 | **88.7** | 66.4 | 89.4 | 65.7 | 62.3 | 71.2 |
| | Dual-RAIL (Xu et al., 2024) | 34.3 | 95.3 | 67.6 | 57.9 | **72.7** | 76.4 | 88.6 | 67.4 | 89.2 | 65.7 | 62.3 | 70.7 |
| | Ours | **38.5** | **95.6** | 74.3 | 60.5 | 72.3 | **84.5** | 88.3 | 68.3 | **89.9** | **66.4** | **64.9** | **73.0** |
| **Last** | LwF (Li & Hoiem, 2017) | 19.4 | 88.6 | 52.6 | 46.8 | 65.9 | 70.1 | 70.3 | 95.2 | 73.9 | 58.5 | 67.1 | 64.4 |
| | iCaRL (Rebuffi et al., 2017) | 26.5 | 86.3 | 47.9 | 50.5 | 55.8 | 66.3 | 70.7 | 93.4 | 85.8 | 60.2 | 66.7 | 64.6 |
| | WiSE-FT (Wortsman et al., 2022) | 30.2 | 93.0 | 59.6 | 58.7 | 73.6 | 82.0 | 79.1 | **97.1** | 86.4 | 65.6 | 68.3 | 72.1 |
| | ZSCL (Zheng et al., 2023) | 24.3 | 91.5 | 72.3 | 59.9 | 74.1 | 86.8 | 87.3 | 93.0 | 88.4 | 71.7 | 71.9 | 74.7 |
| | MoE-Adapter (Yu et al., 2024b) | 32.3 | 95.5 | **76.2** | 68.5 | 40.4 | 95.2 | 87.9 | 60.4 | 89.1 | 66.9 | 69.1 | 71.0 |
| | Primal-RAIL (Xu et al., 2024) | 34.7 | 95.4 | 67.4 | 62.9 | 81.8 | 94.1 | **89.0** | 77.5 | 90.3 | 73.6 | 69.4 | 76.0 |
| | Dual-RAIL (Xu et al., 2024) | 35.0 | 95.3 | 67.5 | 63.7 | **82.7** | 94.8 | 88.7 | 80.6 | 89.9 | 73.7 | 68.9 | 76.4 |
| | Ours | **38.6** | **95.8** | 75.6 | 67.1 | 82.6 | **95.6** | 87.6 | 87.7 | **90.8** | **77.2** | **73.1** | **79.2** |

## 4.2 COMPARISON WITH STATE-OF-THE-ART METHODS

**Cross-domain task-agnostic incremental learning.** Table 1 compares our method with existing baselines in the 16-shot X-TAIL setting across three evaluation metrics. Here, we refer to both under the *Transfer* metric simply as RAIL because Primal- and Dual-RAIL adopt the same domain discriminator. Notably, Dual-RAIL, the current state-of-the-art, stores all historical data representations and replay them during new task training. We consider this a strong but unfair baseline and include it for reference. See Appendix A.7 for more details.

Our method achieves improvements over Primal-RAIL by +0.7%, +1.3%, and +1.6% on *Transfer*, *Average*, and *Last*, respectively. Even compared to the strong baseline Dual-RAIL, we still outperform it by +1.0% on *Average* and +0.5% on *Last*. Additionally, the improvement on *Transfer* indicates that our dynamic aggregation better captures inductive biases from historical tasks and more effectively leverages pre-trained knowledge, rather than relying solely on CLIP's zero-shot capacity. Moreover, CLIP tends to misclassify seen domains

Table 3: Comparison of different CL methods in full-shot X-TAIL setting.

| Method | Transfer | Average | Last | Params. | GPU | Time |
|---|---|---|---|---|---|---|
| WiSE-FT | 45.3 | 55.4 | 75.3 | 149.6M | 32,120MiB | 107m 12s |
| ZSCL | 62.5 | 71.2 | 80.5 | 149.6M | 49,778MiB | 446m 1s |
| MoE-Adapter | 60.1 | 72.5 | 82.7 | 59.8M | 28,486MiB | 151m 55s |
| Primal-RAIL | 64.2 | 73.4 | 82.1 | 24.18M | N/A | **6m 57s** |
| Ours | **64.8** | **75.8** | **87.4** | **4.22M** | 27,948MiB | 25m 21s |

as unseen, while our strategy alleviates this issue, with detailed analysis reported in Appendix A.7.4.

Significant gains are observed in specific domains such as Aircraft, EuroSAT and SUN397, where ours surpass Primal-RAIL by +4.3%, +2.5% and +2.5% on *Average*, respectively. Domains like Aircraft, which involve fine-grained categories (e.g., 707-320), textual descriptions provide limited semantic detail and may be confounded by interference from other classes. In contrast, refined visual prototypes can capture more nuanced semantics, enabling consistent cross-modal alignment, particularly in scenarios where fine-grained distinctions are critical.

Table 3 illustrates the performance of our method in the full-shot X-TAIL setting. Ours achieves notable improvements in *Transfer*, *Average*, and *Last*, outperforming RAIL by +0.6%, +2.4%, and +5.3%, respectively. The contribution of residual prototypes diminishes as more samples improve prototype fidelity (Table 4) but becomes more significant in the 16-shot settings (Table 5). Despite its

higher training time compared to RAIL, due to RAIL's training-free, single-epoch ridge regression design, our method consistently outperforms RAIL in few- and full-shot X-TAIL settings, especially under full-shot conditions where RAIL fails. Lastly, our method uses significantly fewer parameters per task ($4.22M + 515 |\mathcal{C}^t|$ vs. RAIL's $15k |\mathcal{C}^t|$), ensuring greater efficiency when $|\mathcal{C}^t| \geq 4$.

**Multi-Task incremental learning.** Furthermore, in the 5-shot MTIL setting, our method is reduced to using only the prototypes and classes included in the current domain via the domain identity. Comparing with previous baselines, ours also outperforms others, achieving +0.5%, +1.8%, and +3.2% improvements over Primal-RAIL in *Transfer*, *Average*, and *Last*, respectively, and +0.5%, +2.3%, and +2.8% improvements over Dual-RAIL, as shown in Table 2.

### 4.3 ABLATION STUDIES

To better understand the contribution of each component, we conduct a series of ablation studies as shown in Table 4. We begin with a naive continual prompt tuning (PT) baseline and progressively introduce modules. (1) +CA: The cross-attention module (CA) performs static fusion of cross-modal information. (2) +DA: Incorporating the dynamic aggregation (DA) yields performance gains on unseen domains by leveraging knowledge from learned tasks. (3) +RP: Introducing residual prototypes (RP) ensures consistent fidelity and discriminability of prototype. (4) +UaF: Applying uncertainty-aware fusion (UaF) adaptively integrates complementary semantics and promotes unbiased semantic alignment. (5) +EWC: Applying EWC regularization to the CA.

Table 4: Ablation studies on the contribution of each component in full-shot X-TAIL setting.

| Method | DA | RP | UaF | EWC | Transfer | Average | Last |
|---|---|---|---|---|---|---|---|
| CLIP | – | – | – | – | 64.2 | 60.3 | 60.3 |
| w PT | | – | – | – | 64.2 | 73.4 | 84.3 |
| | ✓ | | | | 64.9 | 73.7 | 84.3 |
| w PT&CA | ✓ | | | ✓ | 64.8 | 74.7 | 85.5 |
| | ✓ | ✓ | | ✓ | 64.7 | 74.9 | 85.8 |
| | ✓ | | ✓ | ✓ | 64.9 | 75.3 | 86.5 |
| | ✓ | ✓ | ✓ | | 64.8 | 75.3 | 86.4 |
| | ✓ | ✓ | ✓ | ✓ | 64.8 | 75.8 | 87.4 |

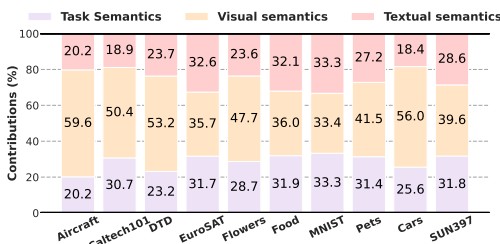

Figure 4: Relative contributions of different semantic objectives in full-shot X-TAIL setting.

As shown in Table 4, DA improves *Transfer* by +0.6%, demonstrating better generalization to unseen domains. Independently, RP improves *Last* by +0.3%, while UaF improves *Last* by +1.0%. Together, they synergistically achieve a 1.9% improvement on *Last*, an additional 0.6% gain, proving their interdependence. Moreover, even without employing EWC, our method maintains a strong advantage, outperforming Primal-RAIL on *Last* by 4.3%. In the 16-shot setting, the benefit of EWC further diminishes (+0.1%, Table 5), indicating that our method inherently suffers little from catastrophic forgetting. Figure 4 illustrates the relative contributions of different semantics across domains. As previously discussed, due to insufficient category-wise semantics in domains like Aircraft, prompt tuning struggles to bridge the modality gap, leading the model to rely more heavily on visual semantics. While domains like Food exhibit richer textual semantics (Figure 3), it still shows a preference for visual semantics. Similar trends are observed across other domains, where prototype semantics contribute more significantly, suggesting its lower uncertainty according to Eq. 13. Although this may be attributed to the residual prototypes, we argue that, visual semantics inherently provide more informative semantics unless textual semantics are carefully crafted.

## 5 CONCLUSION

In this work, we propose a rehearsal-free CL method for CLIP that achieves consistent cross-modal alignment while supporting better generalization to unseen domains. The proposed dynamic aggregation integrates incremental knowledge to enhance CLIP's generalization to unseen domains. Empirical results further suggest that integrating residual prototypes coupled with uncertainty-aware fusion enables more robust modality fusion and effective alignment. Notably, our findings indicate that fine-grained visual semantics provide more stable and informative guidance than textual semantics. Extensive experiments under both MTIL and X-TAIL settings demonstrate that our method achieves state-of-the-art performance while requiring significantly fewer parameters and no replay data.

## ETHICS STATEMENT

This work does not involve new human or animal subjects, personally identifiable data, or sensitive content. All datasets used are publicly available and widely adopted in the community. We believe our method does not raise additional ethical concerns beyond those of existing vision-language continual learning research.

## REPRODUCIBILITY STATEMENT

We have made substantial efforts to ensure the reproducibility of our work. The implementation details of our proposed method, including datasets, training procedures and hyperparameter settings, are provided in Section 4.1 and Appendix A.2. Additional experimental results and ablation studies can be found in Appendix A.7. Theoretical results, including proofs of the main claims, are presented in Appendix A.3, A.4. Furthermore, we provide an anonymous link to the source code and instructions for reproducing all experiments: `https://anonymous.4open.science/r/RP-MSFusion-2777`.

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

# A APPENDIX

## A.1 THE USE OF LARGE LANGUAGE MODELS (LLMS)

We used large language models (LLMs) solely as general-purpose writing assistants to aid with polishing the presentation and improving readability of the paper. LLMs were not involved in research ideation, experimental design, implementation, or result analysis. All technical contributions, experiments, and interpretations are the sole responsibility of the authors.

## A.2 MORE EXPERIMENTAL DETAILS

### A.2.1 DATASETS

In both the X-TAIL and MTIL settings, Order-I follows the alphabetical order of the dataset names. The prompt templates for each dataset are designed as follows:

- Aircraft (Maji et al., 2013): *a photo of a {}, a type of aircraft.*
- Caltech101 (Fei-Fei et al., 2004): *a photo of a {}, a type of aircraft.*
- CIFAR100 (Krizhevsky et al., 2009): *a photo of a {}.*
- DTD (Cimpoi et al., 2014): *a photo of a {} texture.*

- EuroSAT (Helber et al., 2019):        *a centered satellite photo of {}.*

- Flowers (Nilsback & Zisserman, 2008):        *a photo of a {}, a type of flower.*

- Food (Bossard et al., 2014):        *a photo of a {}, a type of food.*

- MNIST (Deng, 2012):        *a photo of the number: {}.*

- OxfordPet (Parkhi et al., 2012):        *a photo of a {}, a type of pet.*

- StanfordCars (Krause et al., 2013):        *a photo of a {}, a type of car.*

- SUN397 (Xiao et al., 2010):        *a photo of a {}.*

In line with (Yu et al., 2024b), Caltech101, EuroSAT, and SUN397 are split into training and test sets using a random seed of 42, following an 80/20 ratio. For all other datasets, we directly utilize the default train/test splits provided by the official PyTorch library. Additionally, for the X-TAIL setting, we adopt the same random order as in (Xu et al., 2024), referred to as Order-II: StanfordCars, Aircraft, OxfordPet, Food, SUN397, MNIST, Flowers, DTD, Caltech101, and EuroSAT.

### A.2.2 EVALUATION METRICS

We provide detailed definitions of the three evaluation metrics *Transfer*, *Average*, and *Last*. Let $a_i^j$ denote the accuracy on the $i$-th domain after the $j$-th task has been learned, where $1 \leq i, j \leq N$ and $N$ is the total number of tasks. Then the metrics for domain $i$ are defined as:

$$
\begin{aligned}
\texttt{Transfer}_i &= \frac{1}{i-1} \sum_{j=1}^{i-1} a_i^j, \quad i = 2, 3, \ldots, N \\
\texttt{Average}_i &= \frac{1}{N} \sum_{j=1}^{N} a_i^j, \quad i = 1, 2, \ldots, N \\
\texttt{Last}_i &= a_i^N, \quad i = 1, 2, \ldots, N
\end{aligned}
\tag{14}
$$

### A.2.3 IMPLEMENTATION DETAILS.

Similarly, we adopt CLIP with a ViT-B/16 (Radford et al., 2021b) visual backbone and conduct all experiments on a single NVIDIA A100 40GB GPU. For each experiment, a few-shot training set is constructed by randomly sampling examples from each class using a fixed random seed of 42. The model is trained for 10 epochs using SGD optimizer with a learning rate of 0.015 and a batch size of 64. For prompt tuning, we use a prompt length of 4 and insert prompts into 9 transformer layers. The probability fusion hyperparameter $\alpha$ is set to 0.8. For multi-semantic fusion, we use a single-head attention mechanism, and the fusion prompt length is also set to 4. The EWC coefficient is set to 400, and the Fisher matrix is updated via exponential moving average.

### A.3 THE ALGORITHM

We summarize the training and inference workflows of our method in Algorithm 1 and Algorithm 2, respectively.

---

**Algorithm 1** Training Phase

---

**Require:** Training data $\mathcal{D}^t = \{(x_i^t, y_i^t)\}_{i=1}^{B^t}$ and textual inputs $\mathcal{W}^t = \{w_i\}_{i=1}^{M^t}$ for all seen classes in task $t$; total number of tasks $N_t$; frozen image encoder $f_\theta(\cdot)$, text encoder $g_\phi(\cdot)$

**Ensure:** Optimized prompts $\boldsymbol{P}^t = \{\boldsymbol{P}_I^t, \boldsymbol{P}_T^t, \boldsymbol{P}_F^t\}$, residual prototypes $\boldsymbol{R}^t$, semantic fusion parameters $\boldsymbol{\Phi}$, and uncertainty coefficients $\boldsymbol{\sigma}^t$

1: Initialize the multi-semantic fusion module $\boldsymbol{\Phi}$
2: **for** $t = 1$ to $N_t$ **do**
3:      Extract image features $\boldsymbol{I} = f_\theta(\boldsymbol{X})$ for all $\boldsymbol{X} \in \mathcal{D}^t$
4:      Compute class-wise prototypes $\boldsymbol{Z}^t$ by averaging $\boldsymbol{I}$ within each class
5:      Initialize task-specific prompts $\boldsymbol{P}_I^t, \boldsymbol{P}_T^t, \boldsymbol{P}_F^t$
6:      Initialize residual prototypes $\boldsymbol{R}^t$ and uncertainty coefficients $\boldsymbol{\sigma}^t = \{\sigma_1^t, \sigma_2^t, \sigma_3^t\}$
7:      **for** epoch $= 1$ to $E$ **do**
8:          **for** mini-batch $(\boldsymbol{X}, Y) \in \mathcal{D}^t$ **do**
9:              Compute prompt-conditioned features: $\boldsymbol{I} = f_\theta(\boldsymbol{X}; \boldsymbol{P}_I^t), \boldsymbol{T} = g_\phi(\mathcal{W}^t; \boldsymbol{P}_T^t)$
10:             Compensate prototypes: $\boldsymbol{Z}^{(1:t)} \leftarrow \boldsymbol{Z}^{(1:t)} + \boldsymbol{R}^{(1:t)}$          ▷ Eq. 8
11:             Compute enhanced features $\hat{\boldsymbol{I}}, \hat{\boldsymbol{T}}, \hat{\boldsymbol{Z}}$ from $\boldsymbol{I}, \boldsymbol{T}, \boldsymbol{Z}^{(1:t)}, \boldsymbol{P}_F^{(1:t)}$, and $\boldsymbol{\Phi}$    ▷ Eq. 9
12:             Obtain multi-semantic probability: $p_{\text{pt}}, p_{\text{it}}, p_{\text{ip}}$         ▷ Eq. 3, 10
13:             Compute individual objectives $\mathcal{L}_1, \mathcal{L}_2, \mathcal{L}_3$            ▷ Eq. 11
14:             Compute total loss $\mathcal{L}_{\text{total}}$ with uncertainty-aware weighting $\boldsymbol{\sigma}^t$     ▷ Eq. 12
15:             Compute final loss: $\mathcal{L} = \mathcal{L}_{\text{total}} + \lambda \cdot \mathcal{L}_{\text{EWC}}$          ▷ Eq. 12
16:             Update $\boldsymbol{P}^t, \boldsymbol{R}^t, \boldsymbol{\sigma}^t$, and $\boldsymbol{\Phi}$ via gradient descent
17:          **end for**
18:      **end for**
19:      Update Fisher Information using EMA
20: **end for**

---

**Algorithm 2** Inference phase

---

**Require:** Test data $\mathcal{D}^{\text{test}}$ and textual inputs $\mathcal{W} = \{w_i\}_{i=1}^M$ for all classes; frozen image encoder $f_\theta(\cdot)$, text encoder $g_\phi(\cdot)$; current trained parameters $\boldsymbol{P}^{(1:t)} = \left\{ \boldsymbol{P}_I^{(1:t)}, \boldsymbol{P}_T^{(1:t)}, \boldsymbol{P}_F^{(1:t)} \right\}, \boldsymbol{R}^{(1:t)}, \boldsymbol{\Phi}$, and $\boldsymbol{\sigma}^{(1:t)}$; current calculated prototypes $\boldsymbol{Z}^{(1:t)}$

1: **for** $x \in \mathcal{D}^{\text{test}}$ **do**
2:      Extract image feature $\boldsymbol{I} = f_\theta(x)$, text features $\boldsymbol{T} = g_\phi(\mathcal{W})$
3:      Compute initial prediction $\hat{y} = \arg\max p_{\text{zs}}(x)$
4:      Compute similarity matrix $\mathcal{S}$ between $\boldsymbol{I}$ and $\boldsymbol{Z}^{(1:t)}$
5:      **if** $\hat{y} \in \mathcal{C}^{seen}$ **then**
6:          Determine the most likely task $\hat{t}$ via $\mathcal{S}$              ▷ Eq. 4
7:          Compute prompt-conditioned features: $\boldsymbol{I} = f_\theta(\boldsymbol{X}; \boldsymbol{P}_I^{\hat{t}}), \boldsymbol{T} = g_\phi(\mathcal{W}^t; \boldsymbol{P}_T^{\hat{t}})$
8:          Compensate prototypes: $\boldsymbol{Z}^{(1:t)} \leftarrow \boldsymbol{Z}^{(1:t)} + \boldsymbol{R}^{(1:t)}$         ▷ Eq. 8
9:          Compute enhanced features $\hat{\boldsymbol{I}}, \hat{\boldsymbol{T}}, \hat{\boldsymbol{Z}}$ from $\boldsymbol{I}, \boldsymbol{T}, \boldsymbol{Z}^{(1:t)}, \boldsymbol{P}_F^{(1:t)}$, and $\boldsymbol{\Phi}$    ▷ Eq. 9
10:          Compute final prediction with $\boldsymbol{\sigma}^{\hat{t}}$            ▷ Eq. 13
11:      **else**
12:          Aggregate historical prompts $\hat{\boldsymbol{P}}$ via $\mathcal{S}$           ▷ Eq. 5
13:          Compute prompt-conditioned features: $\boldsymbol{I} = f_\theta(\boldsymbol{X}; \hat{\boldsymbol{P}}_I), \boldsymbol{T} = g_\phi(\mathcal{W}; \hat{\boldsymbol{P}}_T)$
14:          Calculate final prediction via probability fusion      ▷ Eq. 6
15:      **end if**
16: **end for**

---

## A.4 PROOF OF THEOREMS

### A.4.1 PROOF OF THEOREM 1

*Proof.* The CLIP model aligns visual and textual representations through contrastive learning by jointly optimizing image and text encoders. In this discussion, we focus specifically on the CLIP and, for simplicity, assume that the prompt $\boldsymbol{P}$ are applied solely on the text encoder side.

**Binary classification.** For an input image feature $\boldsymbol{v} \in \mathbb{R}^d$ and a prompt $\boldsymbol{P}$, the text encoder generates a text feature $t(\boldsymbol{P}) \in \mathbb{R}^d$. The output probability is:

$$p(x; \boldsymbol{P}) = \sigma\left(\boldsymbol{v}^\top t(\boldsymbol{P})\right),$$

where $\sigma(z) = \frac{1}{1+e^{-z}}$ is the sigmoid function.

In prompt tuning, $t(\boldsymbol{P})$ is linear or approximately linear:

$$t(\boldsymbol{P}) \approx \boldsymbol{W}_t \boldsymbol{P} + \boldsymbol{b}_t,$$

where $\boldsymbol{W}_t \in \mathbb{R}^{d \times m}$ and $\boldsymbol{b}_t \in \mathbb{R}^d$. Then the logits become:

$$z = \boldsymbol{v}^\top t(\boldsymbol{P}) \approx \boldsymbol{w}^\top \boldsymbol{P} + c,$$

with $\boldsymbol{w} = \boldsymbol{W}_t^\top \boldsymbol{v}$ and $c = \boldsymbol{v}^\top \boldsymbol{b}_t$. The second derivative of the sigmoid function $\sigma(z)$ is:

$$\sigma''(z) = \sigma(z)(1 - \sigma(z))(1 - 2\sigma(z)).$$

In practice, the logit $z$ in CLIP are typically positive due to the use of cosine similarity and a learnable temperature parameter, which amplifies similarity between image and text embeddings. Thus, $p(x; \boldsymbol{P})$ is approximately concave in $\boldsymbol{P}$ within this regime.

**Multi-class classification.** In this task, the probability of class $c$ is given by:

$$p_c(x; \boldsymbol{P}) = \frac{e^{z_c}}{\sum_{j=1}^K e^{z_j}}$$

where $z_j = \boldsymbol{v} t_j(\boldsymbol{P})$ and $t_j(\boldsymbol{P})$ is the text feature for class $j$ from the prompt $\boldsymbol{P}$.

The gradient of $p_c$ is:

$$\nabla_{\boldsymbol{P}} p_c = \sum_{k=1}^K \frac{\partial p_c}{\partial z_k} \frac{\partial z_k}{\partial \boldsymbol{P}}$$

First-order derivatives of softmax:

$$\frac{\partial p_c}{\partial z_k} = \begin{cases} p_c(1 - p_c), & k = c \\ -p_c p_k, & k \neq c \end{cases}$$

Derivative of $z_k$ w.r.t. $\boldsymbol{P}$:

$$\frac{\partial z_k}{\partial \boldsymbol{P}} = \boldsymbol{w}_k, \quad \boldsymbol{w}_k = \boldsymbol{W}_{t,k}^\top \boldsymbol{v}.$$

Final gradient:

$$\nabla_{\boldsymbol{P}} p_c = p_c \left( \boldsymbol{w}_c - \sum_{k=1}^K p_k \boldsymbol{w}_k \right) = p_c(\boldsymbol{w}_c - \bar{\boldsymbol{w}}),$$

where $\bar{\boldsymbol{w}} = \sum_{k=1}^K p_k \boldsymbol{w}_k$. Now, we compute the Hessian matrix $\nabla_{\boldsymbol{P}}^2 p_c \in \mathbb{R}^{m \times m}$:

$$\nabla_{\boldsymbol{P}}^2 p_c = \frac{\partial}{\partial \boldsymbol{P}} \left( \nabla_{\boldsymbol{P}} p_c \right)^\top.$$

Differentiate each term:

$$\nabla_{\boldsymbol{P}}^2 (p_c \boldsymbol{w}_c) = \boldsymbol{w}_c \cdot \nabla_{\boldsymbol{P}} p_c^\top = \boldsymbol{w}_c \cdot \left[ p_c (\boldsymbol{w}_c - \bar{\boldsymbol{w}}) \right]^\top,$$

$$\nabla_{\boldsymbol{P}}^2 \left( \sum_{k=1}^K p_c p_k \boldsymbol{w}_k \right) = \sum_{k=1}^K \left[ p_k \nabla_{\boldsymbol{P}} p_c + p_c \nabla_{\boldsymbol{P}} p_k \right] \cdot \boldsymbol{w}_k^\top.$$

Hence,

$$\nabla_{\boldsymbol{P}}^2 p_c = \boldsymbol{w}_c \cdot [p_c(\boldsymbol{w}_c - \bar{\boldsymbol{w}})]^\top - \sum_{k=1}^K [p_k \nabla_{\boldsymbol{P}} p_c + p_c \nabla_{\boldsymbol{P}} p_k] \cdot \boldsymbol{w}_k^\top.$$

When the CLIP is highly confident in class $c$, i.e., $p_c \approx 1$, $p_k \approx 0$ for $k \neq c$. Then:

$$\nabla_{\boldsymbol{P}}^2 p_c \approx - \sum_{k \neq c} p_c p_k \boldsymbol{w}_k \boldsymbol{w}_k^\top \preceq 0.$$

$\nabla_{\boldsymbol{P}}^2 p_c$ is negative semi-definite and $p_c(x; \boldsymbol{P})$ is locally concave in $\boldsymbol{P}$.

Now, $p(x; \boldsymbol{P})$ is concave in $\boldsymbol{P}$, with weights $\varphi^i \geq 0$ and $\sum_i \varphi^i = 1$. By Jensen's inequality:

$$p\left(x; \hat{\boldsymbol{P}}\right) \geq \sum_i \varphi^i \cdot p\left(x; \boldsymbol{P}^i\right) \geq \varphi^j \cdot p\left(x; \boldsymbol{P}^j\right), \quad j = \mathrm{argmax}_{j \in \{1,\ldots,t\}} \left\{\varphi^j\right\}.$$

Note that:

$$\mathrm{argmax}_c \, p(x; \boldsymbol{P}^j) = \mathrm{argmax}_c \, \varphi^j p(x; \boldsymbol{P}^j).$$

Thus, weighted aggregation outperforms single prompt in expectation. $\qquad\square$

### A.4.2 PROOF OF THEOREM 2

*Proof.* In general, it is assumed that the training and test data are drawn from the same underlying distribution, implying that the noise characteristics observed during both phases remain consistent. Now, suppose there exists a ground-truth logit value $z \sim \mathcal{N}(\mu_0, \sigma_0^2)$, and each semantic objective $k$ observes a noisy version $z_k$ of this true value:

$$z_k = z + \epsilon_k, \quad \epsilon_k \sim \mathcal{N}(0, \sigma_k^2).$$

where each $z_k \sim \mathcal{N}(z, \sigma_k^2)$ is independently observed. By Bayesian rule, the posterior is:

$$p(z \mid z_1, \ldots, z_K) \propto p(z) \prod_{k=1}^K p(z_k \mid z).$$

Substituting the normal distribution forms:

$$p(z) = \frac{1}{\sqrt{2\pi}\sigma_0} \exp\left(-\frac{(z - \mu_0)^2}{2\sigma_0^2}\right), \quad p(z_k \mid z) = \frac{1}{\sqrt{2\pi}\sigma_k} \exp\left(-\frac{(z_k - z)^2}{2\sigma_k^2}\right).$$

To find the MAP estimate $z$, one minimizes the negative log posterior:

$$-\log p(z \mid z_1, \ldots, z_K) = \frac{(z - \mu_0)^2}{2\sigma_0^2} + \sum_{k=1}^K \frac{(z_k - z)^2}{2\sigma_k^2} - \text{constant},$$

$$J = \frac{(z - \mu_0)^2}{2\sigma_0^2} + \sum_{k=1}^K \frac{(z_k - z)^2}{2\sigma_k^2}.$$

Let us expand and gather the terms in $z$:

$$J(z) = \left(\frac{1}{\sigma_0^2} + \sum_{k=1}^K \frac{1}{\sigma_k^2}\right) z^2 - 2\left(\frac{\mu_0}{\sigma_0^2} + \sum_{k=1}^K \frac{z_k}{\sigma_k^2}\right) z + \left(\frac{\mu_0^2}{\sigma_0^2} + \sum_{k=1}^K \frac{z_k^2}{\sigma_k^2}\right).$$

Taking the derivative w.r.t. $z$ and setting it to zero:

$$\frac{\partial J(z)}{\partial z} = 2\left(\frac{1}{\sigma_0^2} + \sum_{k=1}^K \frac{1}{\sigma_k^2}\right) z - 2\left(\frac{\mu_0}{\sigma_0^2} + \sum_{k=1}^K \frac{z_k}{\sigma_k^2}\right) = 0,$$

solving for $\hat{z}$

$$\left(\frac{1}{\sigma_0^2} + \sum_{k=1}^K \frac{1}{\sigma_k^2}\right) \hat{z} = \frac{\mu_0}{\sigma_0^2} + \sum_{k=1}^K \frac{z_k}{\sigma_k^2}.$$

Hence,
$$\hat{z} = \frac{\mu_0/\sigma_0^2 + \sum_{k=1}^{K} z_k/\sigma_k^2}{1/\sigma_0^2 + \sum_{k=1}^{K} 1/\sigma_k^2}.$$

Under a uniform prior $p(z) \propto 1$, i.e., $\sigma_0^2 \to \infty$, the posterior reduces to the likelihood:

$$p(z \mid z_1, \ldots, z_K) \propto \prod_{k=1}^{K} p(z_k \mid z).$$

The term $\mu_0/\sigma_0^2$ and $1/\sigma_0^2$ vanish, producing:

$$\hat{z} = \frac{\sum_{k=1}^{K} z_k/\sigma_k^2}{\sum_{k=1}^{K} 1/\sigma_k^2} \propto \sum_{k=1}^{K} \frac{z_k}{\sigma_k^2}.$$

Thus, the theorem is proved. $\qquad\square$

### A.5 EFFECTIVE OF PROTOTYPE-BASED TASK SELECTION

We leverage class-wise image prototypes to model the task-wise distributional space. As illustrated in Figure 5, we report the prototype-based task selection accuracy as training progresses. Despite the limited 16-shot setting used to compute prototypes, it still achieves remarkably high selection accuracy on most domains, reaching up to 99.0% or even 100.0% (e.g., MNIST). This demonstrates the strong capability of our approach to preserve task-specific distributional characteristics over time, which is critical for accurate task identification in CL.

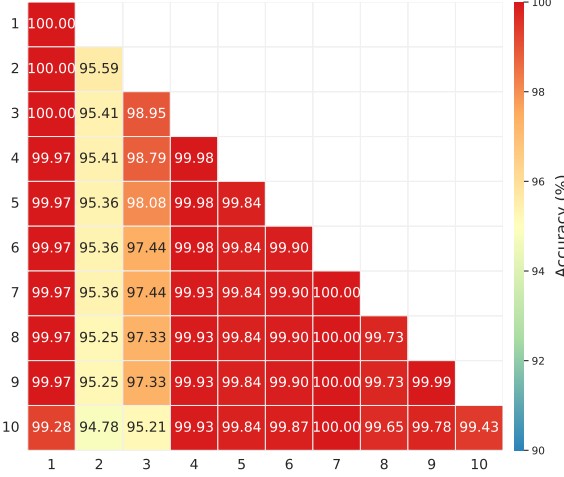

Figure 5: Task selection accuracy based on prototypes across domains during training.

### A.6 EFFECTIVE OF BAYESIAN UNCERTAINTY-AWARE SEMANTIC FUSION

Figure 6 illustrates the relative contributions of different semantics in the 16-shot X-TAIL setting. Combined with Figure 4, we observe that with increased training data, the model learns more reliable vision-language alignment patterns.

### A.7 MORE COMPARISON RESULTS

#### A.7.1 COMPARISON OF DIFFERENT CL METHODS

Table 6 compares our method with existing baselines in the 16-shot X-TAIL setting with order-II across three evaluation metrics. Ours consistently outperforms prior methods, including the strong baseline Dual-RAIL.

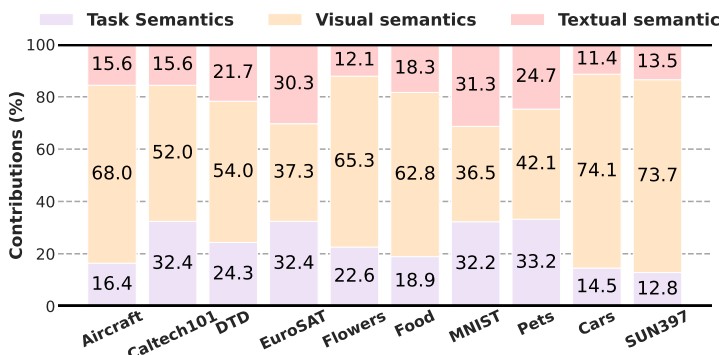

Figure 6: Relative contributions of different semantic objectives on 16-shot X-TAIL.

Table 7 presents a detailed comparison in the full-shot X-TAIL setting with order-I. Primal-RAIL fails to generalize effectively in this setting. In contrast, our method consistently outperforms it in terms of the *Average* score across all domains. Similarly, significant improvements are observed in both *Average* and *Last* metrics. Notably, on the Aircraft, DTD, and EuroSAT domains, our method surpasses Primal-RAIL by 6.7%, 6.0%, and 2.7% in *Average*, and by 12.7%, 15.3%, and 4.7% in *Last*, respectively. Dual-RAIL is omitted due to its substantial computational overhead, as it stores all historical representations and replays them during training. This underscores the efficiency and scalability of our lightweight design in complex CL scenarios.

Table 8, 9, 10 and 11 report the per-domain performance of our method under different settings. In each table, the diagonal entries represent the performance during the learning of the corresponding task, the upper-diagonal entries correspond to the model's generalization performance on unseen domains, and the lower-diagonal entries reflect the performance on previously seen domains, measuring the model's ability to retain knowledge across tasks. Beyond this, We visualize the change in accuracy across all domains as the progresses through different learning stages, as shown in Figure 9.

Table 5: Ablation studies on the contribution of each component on 16-shot X-TAIL.

| Method | DA | RP | UaF | EWC | Transfer | Average | Last |
|---|---|---|---|---|---|---|---|
| CLIP | – | – | – | – | 64.2 | 60.3 | 60.3 |
| w PT | | – | – | – | 64.2 | 70.6 | 79.4 |
| | ✓ | – | – | – | 64.9 | 71.2 | 79.4 |
| w PT&CA | ✓ | | | ✓ | 64.9 | 73.0 | 82.1 |
| | ✓ | ✓ | | ✓ | 64.9 | 73.5 | 83.0 |
| | ✓ | | ✓ | ✓ | 64.8 | 73.2 | 82.3 |
| | ✓ | ✓ | ✓ | | 64.9 | 73.7 | 83.2 |
| | ✓ | ✓ | ✓ | ✓ | 64.9 | 73.7 | 83.3 |

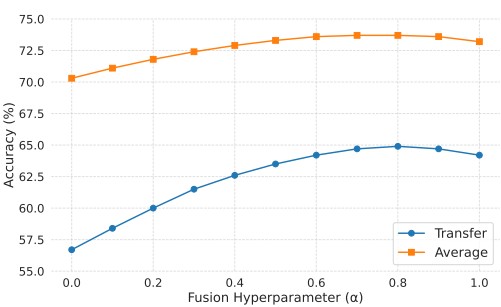

Figure 7: Fusion hyperparameter vs. *Transfer* and *Average* (%) across all domains.

### A.7.2 FUSION HYPERPARAMETER

Figure 7 illustrates the impact of the fusion hyperparameter $\alpha$ on the *Transfer* and *Average*. The best performance is observed at $\alpha = 0.8$, indicating an optimal balance between pre-trained and incremental knowledge. A lower $\alpha$ may underutilize the generalization ability of the pre-trained model, while a higher $\alpha$ could overemphasize pre-trained representations, both leading to suboptimal alignment. These results highlight the importance of appropriately weighting prior and accumulated knowledge to enhance cross-domain generalization.

### A.7.3 PROMPT LENGTH AND INSERTED LAYERS

We evaluate the impact of different prompt length and the number of inserted transformer layers on overall performance, as illustrated in Figure 8. Specifically, we report the averaged results of

*Transfer*, *Average*, and *Last* under each setting. Our default setting adopts a prompt length of 4 and prompts inserted into 9 layers, which achieves the best performance on the *Last*. While a larger configuration (prompt length of 6 with 12 inserted layers) yields better overall results, it incurs higher computational overhead.

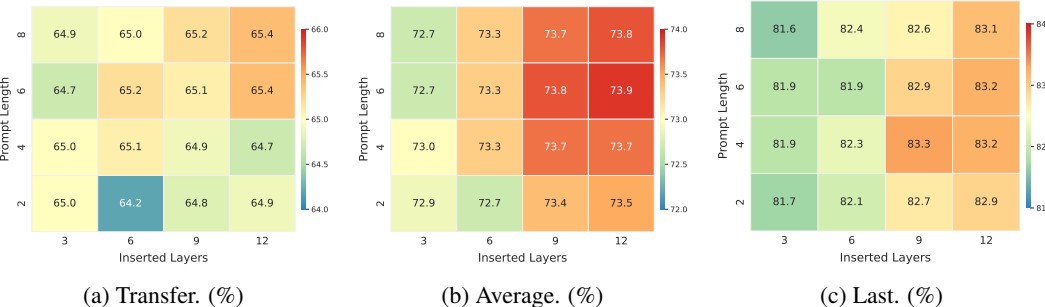

(a) Transfer. (%)      (b) Average. (%)      (c) Last. (%)

Figure 8: Averaged Accuracy (%) across metrics under different prompt lengths and inserted layers.

### A.7.4 MORE ABLATION STUDIES

We also conduct ablation studies in the 16-shot X-TAIL setting to complement earlier experiments as shown in Table 5. Consistent with the results in Section 4.3, the effectiveness of our method is further validated. Besides, EWC provides only a marginal gain (+0.1%) in the few-shot setting. This suggests that our design is inherently stable against forgetting.

Notably, comparing Table 8a and Table 8b shows that the dynamic aggregation (DA) enhances domain discrimination during inference by leveraging accumulated knowledge from seen tasks. Specifically, this improves CLIP's ability to distinguish seen domains such as Caltech101 and DTD, increasing accuracy from 83.8% to 86.8% and from 63.0% to 64.2%, respectively, thus reducing misclassification of seen domains as unseen. Besides, The fidelity of prototypes improves as data increasing, and the contribution of residual prototypes is correspondingly diminished.

### A.8 LIMITATION

While our method achieves state-of-the-art performance in cross-modal CL, several limitations remain. Our method is primarily designed for classification tasks with CLIP, and its applicability to other paradigms (e.g., generation) remains unexplored. As the number of tasks grows, task-specific prompt tuning introduces modest computational overhead and may lead to redundant learning of similar knowledge. In addition, both domain discrimination and prototype construction rely on CLIP's zero-shot capabilities, which may fail to generalize to entirely novel concepts outside its pretraining scope. Nevertheless, this work demonstrates the adaptability and forgetting-resilience of vision-language models in more complex CL scenarios, encouraging the community to explore lifelong learning in realistic applications.

Table 6: Comparison of different CL methods on 16-shot X-TAIL for each domain with order-II in terms of *Transfer*, *Average*, and *Last* scores (%). The best and second results are highlighted with **bold** and underline styles, respectively. RAIL[*] includes both Primal- and Dual-RAIL, as they adopt the same domain discriminator technique. Dual-RAIL[†] stores and replays all historical data.

| | Method | Cars | Aircraft | Pets | Food | SUN397 | MNIST | Flowers | DTD | Caltech101 | EuroSAT | Average |
|---|---|---|---|---|---|---|---|---|---|---|---|---|
| | Zero-shot | 64.0 | 24.8 | 88.8 | 88.0 | 60.2 | 42.5 | 71.0 | 37.5 | 73.8 | 52.0 | 60.3 |
| **Transfer** | LwF (Li & Hoiem, 2017) | – | 16.2 | 81.2 | 79.5 | 60.1 | 32.6 | 52.1 | 31.2 | 67.5 | 20.1 | 48.9 |
| | iCaRL (Rebuffi et al., 2017) | – | 20.4 | 73.1 | 67.8 | 50.5 | 29.9 | 42.8 | 24.2 | 57.8 | 13.6 | 42.2 |
| | WiSE-FT (Wortsman et al., 2022) | – | 21.0 | 83.8 | 80.8 | 57.1 | 35.0 | 58.2 | 32.7 | 65.4 | 31.0 | 51.7 |
| | ZSCL (Zheng et al., 2023) | – | 22.2 | 86.8 | **88.0** | 62.6 | 32.2 | 69.0 | **39.0** | **74.2** | 43.7 | 57.5 |
| | MoE-Adapter (Yu et al., 2024b) | – | 11.7 | **88.8** | **88.0** | 60.2 | 42.5 | **71.1** | 37.0 | 73.8 | 52.0 | 58.3 |
| | RAIL[*] (Xu et al., 2024) | – | **24.8** | 88.8 | **88.0** | 60.2 | 42.5 | 71.0 | 37.5 | 73.8 | 52.0 | 59.8 |
| | Ours | – | 24.6 | **89.6** | 88.3 | 62.7 | **45.7** | 71.0 | 38.3 | 74.1 | **54.2** | **60.9** |
| **Average** | LwF (Li & Hoiem, 2017) | 43.4 | 35.2 | 73.9 | 75.9 | 65.5 | 65.3 | 65.8 | 37.4 | 72.0 | 25.7 | 56.0 |
| | iCaRL (Rebuffi et al., 2017) | 34.4 | 33.6 | 74.3 | 63.0 | 53.2 | 63.9 | 58.8 | 33.4 | 61.7 | 18.7 | 49.5 |
| | WiSE-FT (Wortsman et al., 2022) | 58.3 | 43.9 | 85.7 | 80.7 | 65.3 | 66.3 | 70.2 | 41.1 | 70.7 | 36.7 | 61.9 |
| | ZSCL (Zheng et al., 2023) | 75.9 | 39.9 | 89.7 | 88.8 | 69.4 | 59.7 | 79.2 | 46.2 | 75.6 | 48.2 | 67.3 |
| | MoE-Adapter (Yu et al., 2024b) | 80.4 | 41.9 | 88.9 | 87.8 | 68.3 | **68.3** | 81.1 | 46.5 | 75.6 | 54.6 | 69.3 |
| | Primal-RAIL (Xu et al., 2024) | 80.9 | 43.1 | **91.8** | **89.3** | 67.9 | 58.6 | 80.9 | 46.5 | 77.9 | 55.5 | 69.2 |
| | Dual-RAIL[†] (Xu et al., 2024) | 82.2 | 44.0 | 91.6 | **89.3** | 68.1 | 59.9 | 81.2 | 47.3 | **78.0** | 55.6 | 69.7 |
| | Ours | **82.8** | **46.2** | 92.7 | 88.2 | 69.7 | 61.7 | 81.4 | 48.2 | 77.9 | **57.9** | **70.7** |
| **Last** | LwF (Li & Hoiem, 2017) | 24.8 | 22.2 | 59.7 | 68.3 | 66.7 | 97.8 | 76.1 | 47.5 | 89.5 | 76.1 | 62.9 |
| | iCaRL (Rebuffi et al., 2017) | 32.0 | 35.5 | 80.2 | 68.0 | 60.8 | **98.1** | 79.0 | 54.2 | 76.7 | 64.7 | 64.9 |
| | WiSE-FT (Wortsman et al., 2022) | 41.5 | 30.3 | 78.1 | 76.6 | 68.9 | 96.3 | 81.6 | 57.9 | 91.6 | 87.8 | 71.1 |
| | ZSCL (Zheng et al., 2023) | 72.3 | 36.0 | 87.5 | 88.9 | 73.3 | 83.6 | 93.6 | 61.5 | 81.0 | 89.1 | 76.7 |
| | MoE-Adapter (Yu et al., 2024b) | 80.5 | 45.3 | 88.9 | 87.7 | 73.7 | 94.2 | 96.2 | 68.8 | 82.7 | 77.9 | 79.6 |
| | Primal-RAIL (Xu et al., 2024) | 80.6 | 44.9 | 92.7 | 90.1 | 73.3 | 91.0 | 95.7 | 67.7 | 94.3 | 86.7 | 81.7 |
| | Dual-RAIL[†] (Xu et al., 2024) | 82.3 | 45.9 | 92.6 | **90.3** | 73.7 | 94.4 | 96.4 | 70.2 | **94.9** | 88.4 | 82.9 |
| | Ours | **82.6** | **48.7** | **93.5** | 88.2 | **74.2** | 92.5 | **97.1** | **71.8** | 93.8 | **90.8** | **83.3** |

Table 7: Comparison of different CL methods on full-shot X-TAIL for each domain with order-I in terms of *Transfer*, *Average*, and *Last* scores (%). The best and second results are highlighted with **bold** and underline styles, respectively. Dual-RAIL is excluded due to its prohibitive computational cost.

| | Method | Aircraft | Caltech101 | DTD | EuroSAT | Flowers | Food | MNIST | Pets | Cars | SUN397 | Average |
|---|---|---|---|---|---|---|---|---|---|---|---|---|
| | Zero-shot | 24.8 | 73.9 | 37.5 | 52.0 | 71.0 | 88.0 | 42.5 | 88.8 | 64.0 | 60.2 | 60.3 |
| **Transfer** | LwF (Li & Hoiem, 2017) | – | 69.9 | 30.2 | 26.2 | 50.6 | 73.1 | 34.0 | 71.6 | 32.1 | 53.2 | 49.0 |
| | iCaRL (Rebuffi et al., 2017) | – | 47.4 | 20.0 | 16.1 | 35.8 | 55.7 | 20.3 | 57.6 | 15.1 | 44.0 | 34.7 |
| | WiSE-FT (Wortsman et al., 2022) | – | 63.4 | 28.9 | 29.3 | 47.8 | 68.6 | 28.9 | 64.1 | 27.2 | 49.7 | 45.3 |
| | ZSCL (Zheng et al., 2023) | – | **76.4** | 37.3 | 46.9 | 68.6 | 87.9 | 35.5 | 86.5 | 59.2 | **64.3** | 62.5 |
| | MoE-Adapter (Yu et al., 2024b) | – | 73.9 | 33.4 | 28.1 | 67.3 | **88.0** | **45.9** | 88.8 | 58.2 | 57.2 | 60.1 |
| | Primal-RAIL (Xu et al., 2024) | – | 73.9 | 37.5 | **52.0** | **71.0** | **88.0** | 42.5 | 88.8 | **64.0** | 60.2 | 64.2 |
| | Ours | – | 75.4 | **37.7** | 51.9 | 70.9 | **88.0** | 43.8 | **89.2** | 63.8 | 62.8 | **64.8** |
| **Average** | LwF (Li & Hoiem, 2017) | 32.7 | 83.9 | 50.0 | 57.2 | 57.4 | 74.3 | 60.2 | 76.5 | 42.4 | 55.9 | 59.1 |
| | iCaRL (Rebuffi et al., 2017) | 36.7 | 69.1 | 45.0 | 50.8 | 53.8 | 60.8 | 52.0 | 67.9 | 28.2 | 47.6 | 51.0 |
| | WiSE-FT (Wortsman et al., 2022) | 23.9 | 83.9 | 47.8 | 48.8 | 56.7 | 71.7 | 57.2 | 72.8 | 38.5 | 52.8 | 55.4 |
| | ZSCL (Zheng et al., 2023) | 46.4 | 79.8 | 58.7 | 77.3 | 82.2 | 89.9 | 58.8 | 88.8 | 64.6 | **65.8** | 71.2 |
| | MoE-Adapter (Yu et al., 2024b) | 51.8 | 89.1 | 62.2 | 72.2 | 84.8 | 88.2 | 65.7 | 88.8 | 62.8 | 59.1 | 72.5 |
| | Primal-RAIL (Xu et al., 2024) | 47.9 | 88.9 | 57.6 | 81.3 | 83.3 | 90.0 | 64.7 | 90.3 | 67.7 | 62.2 | 73.4 |
| | Ours | **54.6** | **89.7** | **63.6** | **84.0** | **85.9** | **90.3** | **66.1** | **90.7** | **68.4** | 64.7 | **75.8** |
| **Last** | LwF (Li & Hoiem, 2017) | 21.8 | 79.2 | 58.1 | 64.8 | 67.0 | 83.6 | **99.5** | 89.2 | 82.1 | 80.2 | 72.5 |
| | iCaRL (Rebuffi et al., 2017) | 42.8 | 77.8 | 60.1 | 71.2 | 79.5 | 83.6 | **99.5** | 92.1 | 78.8 | 79.3 | 76.5 |
| | WiSE-FT (Wortsman et al., 2022) | 30.5 | 87.3 | 57.4 | 63.4 | 74.0 | 88.0 | **99.6** | 92.5 | 80.4 | 80.0 | 75.3 |
| | ZSCL (Zheng et al., 2023) | 39.7 | 81.3 | 62.0 | 77.3 | 90.2 | 91.6 | 95.9 | 93.7 | 85.5 | 78.8 | 80.5 |
| | MoE-Adapter (Yu et al., 2024b) | 51.9 | 90.4 | 68.2 | 90.6 | **96.3** | 88.3 | 95.0 | 88.9 | 81.3 | 76.5 | 82.7 |
| | Primal-RAIL (Xu et al., 2024) | 42.1 | **94.7** | 59.8 | 93.8 | 86.6 | 91.9 | 97.8 | 93.4 | 81.1 | 79.9 | 82.1 |
| | Ours | **54.8** | 94.2 | **75.1** | **98.5** | 96.0 | **92.7** | **99.5** | **94.1** | **86.7** | **82.0** | **87.4** |

Table 8: Accuracy (%) of our method on 16-shot X-TAIL with order-I. Each row reports the model's performance on all domains after training the corresponding task.

(a) Our method with dynamic aggregation (DA).

| | Aircraft | Caltech101 | DTD | EuroSAT | Flowers | Food | MNIST | Pets | Cars | SUN397 | Average |
|---|---|---|---|---|---|---|---|---|---|---|---|
| **Transfer** | | 75.7 | 37.2 | 52.1 | 71.1 | 88.3 | 43.7 | 89.2 | 63.7 | 62.9 | **64.9** |
| Aircraft | **48.9** | 75.7 | 36.8 | 51.7 | 71.2 | 88.4 | 44.6 | 89.5 | 64.5 | 62.3 | |
| Caltech101 | 50.4 | **86.8** | 37.7 | 52.0 | 70.9 | 88.4 | 45.7 | 89.1 | 63.9 | 62.9 | |
| DTD | 49.9 | 87.8 | **64.2** | 52.4 | 71.1 | 88.3 | 42.1 | 89.0 | 63.8 | 62.9 | |
| EuroSAT | 50.0 | 87.6 | 64.1 | **90.2** | 71.2 | 88.2 | 43.1 | 89.0 | 63.8 | 62.6 | |
| Flowers | 50.0 | 89.0 | 65.2 | 89.6 | **97.1** | 88.0 | 42.9 | 89.0 | 63.5 | 62.8 | |
| Food | 49.5 | 89.1 | 65.3 | 89.6 | 97.1 | **88.6** | 43.7 | 89.3 | 63.1 | 62.9 | |
| MNIST | 49.7 | 89.1 | 65.7 | 90.3 | 97.2 | 88.7 | **94.9** | 89.5 | 63.5 | 63.3 | |
| Pets | 49.1 | 90.0 | 65.3 | 90.6 | 97.1 | 88.6 | 94.2 | **93.4** | 63.8 | 63.1 | |
| Cars | 49.1 | 91.3 | 65.6 | 90.2 | 97.0 | 88.7 | 94.1 | 93.7 | **83.1** | 63.0 | |
| **SUN397 (Last)** | 49.1 | 92.9 | 70.0 | 90.7 | 97.2 | 88.5 | 93.9 | 93.5 | 83.0 | **74.2** | **83.3** |
| **Average** | 49.6 | 88.0 | 60.0 | 78.7 | 86.7 | 88.4 | 63.9 | 90.5 | 67.6 | 64.0 | **73.7** |

(b) Our method without dynamic aggregation (w/o DA).

| | Aircraft | Caltech101 | DTD | EuroSAT | Flowers | Food | MNIST | Pets | Cars | SUN397 | Average |
|---|---|---|---|---|---|---|---|---|---|---|---|
| **Transfer** | | 73.9 | 37.5 | 52.0 | 71.0 | 88.0 | 42.5 | 88.8 | 64.0 | 60.2 | **64.2** |
| Aircraft | **48.9** | 73.9 | 37.5 | 52.0 | 71.0 | 88.0 | 42.5 | 88.8 | 64.0 | 60.2 | |
| Caltech101 | 50.4 | **83.8** | 37.5 | 52.0 | 71.0 | 88.0 | 42.5 | 88.8 | 64.0 | 60.2 | |
| DTD | 49.9 | 85.2 | **63.0** | 52.0 | 71.0 | 88.0 | 42.5 | 88.8 | 64.0 | 60.2 | |
| EuroSAT | 50.0 | 85.3 | 63.1 | **90.0** | 71.0 | 88.0 | 42.5 | 88.8 | 64.0 | 60.2 | |
| Flowers102 | 50.0 | 87.3 | 64.1 | 89.4 | **97.1** | 88.0 | 42.5 | 88.8 | 64.0 | 60.2 | |
| Food101 | 49.5 | 87.6 | 64.7 | 89.3 | 97.1 | **88.6** | 42.5 | 88.8 | 64.0 | 60.2 | |
| MNIST | 49.7 | 87.6 | 65.0 | 90.1 | 97.2 | 88.7 | **94.9** | 88.8 | 64.0 | 60.2 | |
| OxfordPets | 49.1 | 89.1 | 64.9 | 90.4 | 97.1 | 88.6 | 94.2 | **93.4** | 64.0 | 60.2 | |
| StanfordCars | 49.1 | 90.5 | 65.1 | 90.0 | 97.0 | 88.7 | 94.1 | 93.7 | **83.1** | 60.2 | |
| **SUN397 (Last)** | 49.1 | 92.9 | 70.0 | 90.7 | 97.2 | 88.5 | 93.9 | 93.5 | 83.0 | **74.2** | **83.3** |
| **Average** | 49.6 | 86.3 | 59.5 | 78.6 | 86.7 | 88.3 | 63.2 | 90.2 | 67.8 | 61.6 | **73.2** |

Table 9: Accuracy (%) of our method on 16-shot X-TAIL with order-II. Each row reports the model's performance on all domains after training the corresponding task.

| | Cars | Aircraft | Pets | Food | SUN397 | MNIST | Flowers | DTD | Caltech101 | EuroSAT | Average |
|---|---|---|---|---|---|---|---|---|---|---|---|
| **Transfer** | | 24.6 | 89.6 | 88.3 | 62.7 | 45.7 | 71.0 | 38.3 | 74.1 | 54.2 | **60.9** |
| Cars | **83.3** | 24.6 | 89.3 | 88.3 | 62.3 | 45.3 | 70.7 | 38.2 | 75.4 | 51.5 | |
| Aircraft | 83.0 | **48.3** | 89.8 | 88.3 | 62.9 | 44.0 | 70.1 | 38.7 | 75.0 | 53.6 | |
| Pets | 83.0 | 48.8 | **93.3** | 88.4 | 63.0 | 47.4 | 71.1 | 38.5 | 73.7 | 54.6 | |
| Food | 82.5 | 48.9 | 93.3 | **88.3** | 62.6 | 45.8 | 71.3 | 37.8 | 74.4 | 55.3 | |
| SUN397 | 82.7 | 48.8 | 93.6 | 88.2 | **74.2** | 46.1 | 71.4 | 38.0 | 73.6 | 54.5 | |
| MNIST | 83.0 | 48.8 | 93.4 | 88.4 | 74.5 | **55.9** | 71.2 | 38.1 | 73.6 | 55.4 | |
| Flowers | 82.7 | 47.8 | 93.7 | 88.1 | 74.4 | 55.8 | **97.1** | 38.8 | 73.4 | 55.6 | |
| DTD | 82.5 | 48.6 | 93.6 | 87.9 | 74.1 | 91.3 | 97.2 | **70.0** | 73.3 | 54.8 | |
| Caltech101 | 82.6 | 48.5 | 93.4 | 88.1 | 74.6 | 93.2 | 97.1 | 71.8 | **93.2** | 52.8 | |
| **EuroSAT (Last)** | 82.6 | 48.7 | 93.5 | 88.2 | 74.2 | 92.5 | 97.1 | 71.8 | 93.8 | **90.8** | **83.3** |
| **Average** | 82.8 | 46.2 | 92.7 | 88.2 | 69.7 | 61.7 | 81.4 | 48.2 | 77.9 | 57.9 | **70.7** |

Table 10: Accuracy (%) of our method on full-shot X-TAIL with order-I. Each row reports the model's performance on all domains after training the corresponding task.

| | Aircraft | Caltech101 | DTD | EuroSAT | Flowers | Food | MNIST | Pets | Cars | SUN397 | Average |
|---|---|---|---|---|---|---|---|---|---|---|---|
| **Transfer** | | 75.4 | 37.7 | 51.9 | 70.9 | 88.0 | 43.8 | 89.2 | 63.8 | 62.8 | **64.8** |
| Aircraft | **54.9** | 75.4 | 37.2 | 51.0 | 71.1 | 88.3 | 45.1 | 89.2 | 64.6 | 61.9 | |
| Caltech101 | 55.0 | **89.8** | 38.2 | 52.6 | 70.9 | 88.2 | 48.9 | 89.4 | 64.2 | 62.7 | |
| DTD | 54.9 | 90.4 | **69.3** | 51.9 | 71.5 | 88.1 | 43.3 | 89.2 | 63.9 | 63.0 | |
| EuroSAT | 54.5 | 89.2 | 67.8 | **97.4** | 70.1 | 87.6 | 42.0 | 88.9 | 62.8 | 62.4 | |
| Flowers | 54.2 | 90.7 | 69.2 | 97.2 | **95.9** | 87.7 | 41.7 | 89.1 | 63.2 | 62.7 | |
| Food | 54.3 | 91.1 | 70.1 | 97.0 | 95.9 | **92.8** | 42.1 | 89.3 | 63.5 | 63.2 | |
| MNIST | 54.5 | 91.0 | 69.8 | 98.3 | 95.8 | 92.7 | **99.4** | 89.2 | 63.9 | 62.9 | |
| Pets | 54.5 | 92.1 | 69.6 | 98.3 | 95.8 | 92.6 | 99.5 | **94.3** | 64.1 | 62.9 | |
| Cars | 54.7 | 93.6 | 69.5 | 98.2 | 96.1 | 92.7 | 99.5 | 94.2 | **87.2** | 63.1 | |
| **SUN397 (Last)** | 54.8 | 94.2 | 75.1 | 98.5 | 96.0 | 92.7 | 99.5 | 94.1 | 86.7 | **82.0** | **87.4** |
| **Average** | 54.6 | 89.7 | 63.6 | 84.0 | 85.9 | 90.3 | 66.1 | 90.7 | 68.4 | 64.7 | **75.8** |

Table 11: Accuracy (%) of our method on 5-shot MTIL with order-I. Each row reports the model's performance on all domains after training the corresponding task.

| | Aircraft | Caltech101 | CIFAR100 | DTD | EuroSAT | Flowers | Food | MNIST | Pets | Cars | SUN397 | Average |
|---|---|---|---|---|---|---|---|---|---|---|---|---|
| **Transfer** | | 94.1 | 69.4 | 43.7 | 55.4 | 71.3 | 89.0 | 57.8 | 89.5 | 64.0 | 64.1 | **69.8** |
| Aircraft | **38.4** | 94.1 | 68.9 | 43.2 | 55.0 | 71.3 | 88.9 | 60.3 | 89.7 | 64.5 | 63.1 | |
| Caltech101 | 38.9 | **95.8** | 70.0 | 43.7 | 54.9 | 71.4 | 89.0 | 59.1 | 89.6 | 64.2 | 64.2 | |
| CIFAR100 | 38.5 | 95.6 | **75.3** | 44.0 | 56.2 | 71.2 | 89.1 | 58.3 | 89.2 | 63.4 | 64.1 | |
| DTD | 38.6 | 95.8 | 75.2 | **66.4** | 55.3 | 71.4 | 89.1 | 56.9 | 89.5 | 63.7 | 64.1 | |
| EuroSAT | 38.3 | 95.8 | 75.1 | 66.6 | **81.8** | 71.2 | 89.1 | 58.7 | 89.4 | 64.3 | 64.2 | |
| Flowers | 38.6 | 95.8 | 75.0 | 66.2 | 82.2 | **95.1** | 89.0 | 55.9 | 89.7 | 64.1 | 64.1 | |
| Food | 38.5 | 95.3 | 75.5 | 66.9 | 81.2 | 95.4 | **87.4** | 55.4 | 89.6 | 63.8 | 64.3 | |
| MNIST | 38.6 | 95.6 | 75.4 | 67.3 | 82.3 | 95.3 | 87.4 | **85.2** | 89.5 | 64.1 | 64.2 | |
| Pets | 38.6 | 96.1 | 76.0 | 66.5 | 82.4 | 95.5 | 87.3 | 86.8 | **90.7** | 63.9 | 64.5 | |
| Cars | 38.1 | 96.1 | 75.7 | 67.1 | 81.7 | 95.6 | 87.8 | 87.1 | 90.9 | **77.2** | 64.5 | |
| **SUN397 (Last)** | 38.6 | 95.8 | 75.6 | 67.1 | 82.6 | 95.6 | 87.6 | 87.7 | 90.8 | 77.2 | **73.1** | **79.2** |
| **Average** | 38.5 | 95.6 | 74.3 | 60.5 | 72.3 | 84.5 | 88.3 | 68.3 | 89.9 | 66.4 | 64.9 | **73.0** |

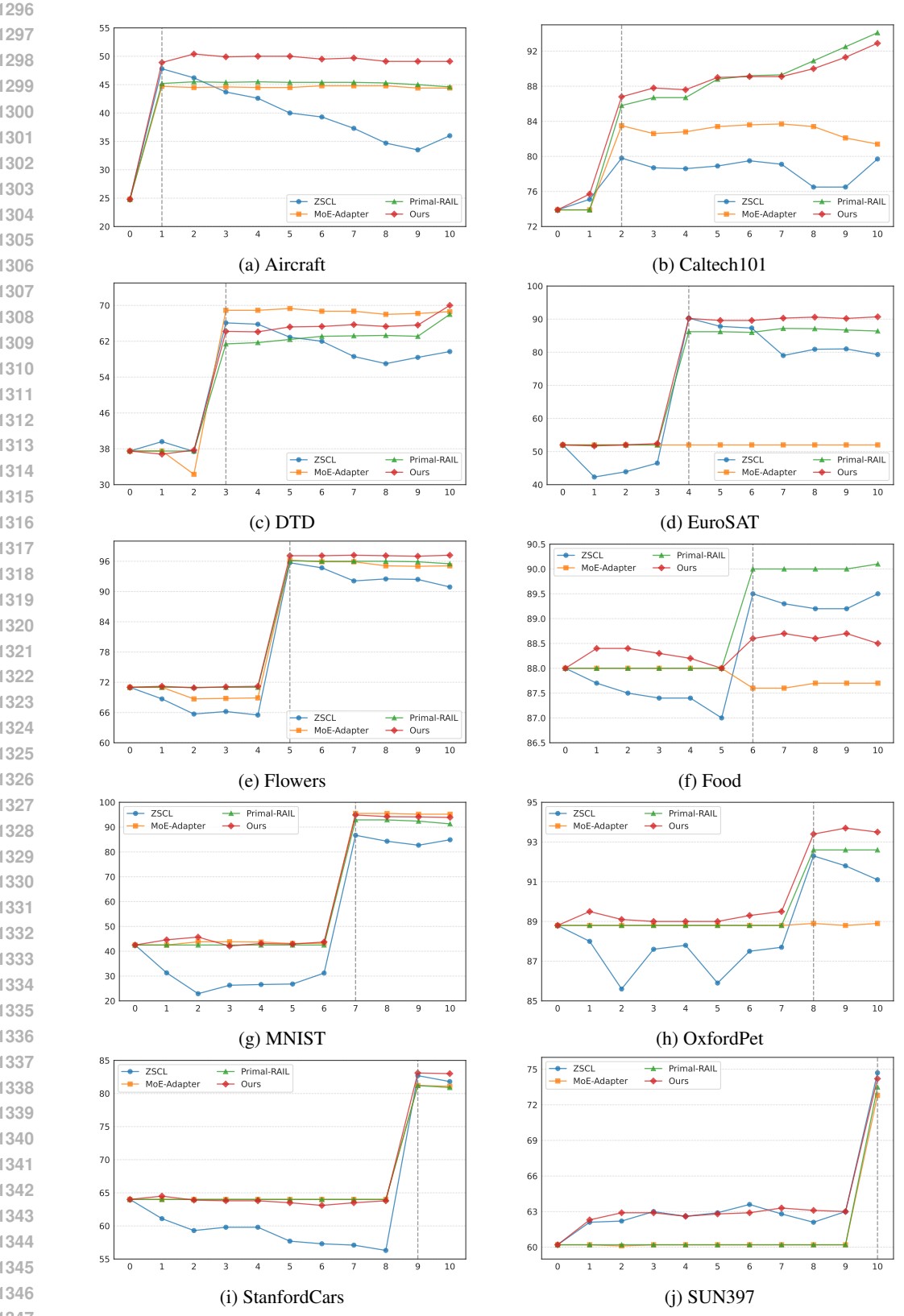

Figure 9: Accuracy (%) on all domains throughout the training process. For example, (a) illustrates the accuracy of Aircraft progression at each learning stage.

