# OpenReview forum: "Towards Consistent Cross-Modal Alignment in Continual Learning for Vision-Language Models"
_ICLR.cc/2026/Conference — ICLR 2026 Conference Withdrawn Submission_

### Official Review · Reviewer_F1Ss · 2025-10-27

**Soundness:** 2
**Presentation:** 1
**Contribution:** 2
**Rating:** 4
**Confidence:** 2

**Summary:**

This paper proposes a new method of continual learning for CLIP-like multi-modal contrastive models. The proposed method consists of four components: 1) task identification (for seen domains) or ensemble (for unseen domains) weighted by cosine similarity to prototype vectors during inference; 2) adapter-tuning for prototype vectors; 3) cross-attention transformation for image/text/prototype vectors; 4) balanced weighting for continual learning losses. Experiments show that the proposed method achieves superior or comparable performance to the existing methods, even without using rehearsal data.

**Strengths:**

- The experimental results suggest that the proposed method works well on cross-domain or multi-domain continual learning scenarios, even without retaining past training data. In some cases the method outperforms the existing methods that retain past training data, but in other cases it performs worse than several previous methods.
- The ablation study shows how each component in the proposed method contributes to the performance gain.

**Weaknesses:**

- The paper is not well-structured; the explanations for the training and inference phases are mixed up, except for the caption of Figure 2, which prevent readers from precise understanding for the overall procedure.
- Their method and evaluation only focus on continual learning for the CLIP architecture, which raises a concern about its limitation of applicability to other multi-modal or vision-language models.
- Theorem 1 claims that inference with the weighted average of adapter parameters yields higher output probability than averaging the output probabilities with the same weighting. However, since model outputs are not calibrated probabilities in general, the practical implication of this claim remains unclear. Moreover, the proof relies on assuming the concavity of model outputs with respect to the adapter's parameter P, which does not hold for actual neural networks. Even if such an assumption was true, the claim is just a direct application of Jensen's inequality.
- Theorem 2 claims the MAP estimation is approximately achieved by a weighted sum of (Gaussian) random variables $z_k$, leading to the derivation of the weighted distribution in eq (13). However, the resulting distribution seems NOT the corresponding distribution for Theorem 2. Indeed, the distribution of a weighted sum of random variables should be the form of convolution, while the distribution in eq (13) is just a variant of the mixture distribution. Could you clarify the relationship between Theorem 2 and the distribution in eq (13)?

**Questions:**

See the above weaknesses.

---

### Official Review · Reviewer_sVeE · 2025-10-29

**Soundness:** 2
**Presentation:** 2
**Contribution:** 2
**Rating:** 2
**Confidence:** 3

**Summary:**

This paper addresses continual learning for vision-language models (VLMs) pre-trained on large-scale text-image pair datasets like CLIP. Existing baselines prevent catastrophic forgetting by distilling from pre-trained models or utilizing rehearsal that preserves past data, but this incurs significant computational overhead. Recent approaches introduce adapters for additional training or class-specific prototypes to prevent knowledge interference between tasks and efficiently adapt models. The paper specifically argues that prototype-based methods suffer from insufficient prototype fidelity and separability, leading to suboptimal performance. To address this challenge, the paper proposes a rehearsal-free continual learning method that incorporates (i) improved fidelity in residual prototypes and (ii) dynamic fusion of visual/textual semantics via uncertainty estimation. Experiments report that the proposed method achieves competitive performance compared to rehearsal-based baselines.

**Strengths:**

- **S1.** The paper proposes a novel method for improving prototype-based VLM continuous learning.
- **S2.** The paper confirms the effectiveness of the proposed method through standard sequential learning benchmarks and ablation studies.

**Weaknesses:**

- **W1.** There is insufficient comparison with the prototype baseline. In particular, comparison with LADA [a], which is directly related to the proposed method, is essential. When compared with the X-TAIL results reported by [a], little difference is observed with the proposed method, which may suggest that the paper is solving a problem that is not particularly significant in continuous learning for VLMs.
- **W2.** The verification of the research question is insufficient. The paper does not clearly define fidelity and separability, instead evaluating them using task performance as a proxy. To confirm the paper's claims, these definitions must be clarified, and an analysis must be conducted to determine the actual extent of improvement achieved by the proposed method. Furthermore, the paper should clarify whether the estimated uncertainty truly captures the semantic reliability of the modality, or if it merely balances the loss according to task difficulty.
- **W3.** The differences from existing prototype bases and the positioning of the proposed method have not been discussed accurately. Section 2 merely lists existing research without explicitly discussing how this paper relates to it or where its novelty lies.
- **W4.** No validation has been performed outside of classification tasks. If the proposed method improves semantic alignment between images and text, its effectiveness should be greater in cross-modal search tasks than in classification using short text.

[a] Luo, Mao-Lin, et al. "LADA: Scalable Label-Specific CLIP Adapter for Continual Learning." ICML2025.

**Questions:**

Please response the concerns raised in the weaknesses section.

---

### Official Review · Reviewer_6Bgd · 2025-10-29

**Soundness:** 3
**Presentation:** 2
**Contribution:** 2
**Rating:** 4
**Confidence:** 4

**Summary:**

This paper proposes a novel continual learning method for CLIP.
The proposed method addresses the problems of conventional continual learning methods: reliability of prototypes and modality alignment
by dynamically adjusting mean prototypes of visual embeddings and text embeddings via prompt tuning.
Experimental results demonstrate that the proposed method alleviates catastrophic forgetting and improves accuracy in continual classification tasks.

**Strengths:**

- S1: The experiments are well conducted.
- S2: Addressing the uncertainty of prototypes is interesting.

**Weaknesses:**

- W1: The definitions of some terms, e.g., seen and unseen, domain and task, are ambiguous, which makes understanding of the proposed method harder. They should be clarified in the problem setting.
- W2: During inference, according to Sec. 3.2, the proposed method first identifies the input's domain. However, the procedure of domain identification is not described (as well as the definition of domain).
- W3: The motivation behind the design of the task-shared cross-attention is not clear. What problem does this module address? Ablation on this module would also be helpful.
- W4: Performance improvements by the proposed method are marginal; most improvements compared to the second baselines are less than 1%pt.
- W5: Since the proposed method addresses the modality gap, directly evaluating and visualizing it would be beneficial. For example, computing pairwise similarities or visualizing the embedding space, like [a] and [b], would be interesting.
- W6 (minor): Is Theorem 1 misplaced? It should be introduced after Eq. (5).



[a] Eslami and de Melo, Mitigate the Gap: Investigating Approaches for Improving Cross-Modal Alignment in CLIP, ICLR 2025.
[b] Liang et al., Mind the gap: Understanding the modality gap in multi-modal contrastive representation learning, NeurIPS 2022.

**Questions:**

Q1: How were the relative contributions in Fig. 4 computed?

---

### Official Review · Reviewer_zsqA · 2025-11-11

**Soundness:** 1
**Presentation:** 2
**Contribution:** 2
**Rating:** 2
**Confidence:** 4

**Summary:**

This paper proposes a multimodal few-shot learning framework integrating prompt augmentation, residual prototype refinement, and uncertainty-weighted fusion. The authors present Theorem 1, claiming theoretical advantages of weighted aggregation, and demonstrate empirical improvements on benchmark datasets. However, the work primarily combines existing techniques — residual connections from ResNet and uncertainty weighting from multi-task learning — without substantial innovation. Critical theoretical claims lack validation and rest on unverified assumptions (e.g., concavity of output probabilities in deep networks).

**Strengths:**

1. The paper addresses a practically relevant problem of multimodal few-shot learning with a well-structured framework that systematically combines multiple components.

2. Comprehensive experimental validation across multiple benchmarks demonstrates consistent performance improvements, with thorough ablation studies showing the contribution of each module.

3. The writing is generally clear and well-organized, with effective use of figures and tables to illustrate the proposed approach and experimental results.

**Weaknesses:**

1. Theorem 1 lacks empirical validation and rests on invalid assumptions. The claim that weighted aggregation outperforms any single prompt assumes concavity of output probabilities with respect to prompt parameters, which typically does not hold in deep neural networks. The paper provides no empirical validation of this assumption. Table 4 only contrasts "with DA" versus "without DA," failing to verify Theorem 1's central claim through direct comparison between weighted aggregation and individual prompts. Moreover, when a significantly superior prompt exists, weighted aggregation may degrade performance by incorporating suboptimal prompts, potentially underperforming direct use of the superior prompt alone, and contradicting the theoretical assertion.

2. The residual prototype offers no methodological novelty. This component directly transplants ResNet's residual connection to class prototypes ($Z^{\text{aug}} = Z + R$), representing a mere engineering application of standard techniques from model architecture to prototype space. The authors neither justify why the residual form outperforms alternative augmentation schemes (e.g., attention-based weighting) nor provide a theoretical analysis explaining how it mitigates prototype interference. While ablation studies show effectiveness, this likely stems from increased model parameters rather than the inherent superiority of the residual design.

3. The uncertainty weighting mechanism lacks originality. Equation 12 directly adopts the homoscedastic uncertainty framework from [1], including the loss formulation $\mathcal{L} = \sum [1/\sigma_k^2 \cdot \mathcal{L}_k + \log \sigma_k]$ and theoretical derivation, merely transferring this technique from multi-task learning to multimodal fusion without modifications or extensions. While possessing engineering value, this constitutes no methodological contribution. The authors neither discuss the method's suitability for the current scenario nor compare it with alternative uncertainty estimation methods (e.g., evidential deep learning [2] or Monte Carlo dropout [3]).

[1] Multi-task learning using uncertainty to weigh losses for scene geometry and semantics. CVPR 2018.
[2] Evidential deep learning to quantify classification uncertainty. NeurIPS 2018.
[3] Dropout as a Bayesian approximation: Representing model uncertainty in deep learning. ICML 2016.

**Questions:**

1. Can the authors provide empirical evidence supporting the concavity assumption in Theorem 1, and conduct experiments directly comparing weighted aggregation against each prompt?

2. What is the theoretical justification for using residual connections in prototype space? Have the authors considered alternative augmentation designs?

3. Why was Kendall et al.'s uncertainty weighting chosen for this multimodal scenario? How does it compare with other uncertainty estimation methods, such as evidential deep learning or Monte Carlo dropout, in this context?

---

### Note · Authors · 2025-12-04

I have read and agree with the venue's withdrawal policy on behalf of myself and my co-authors.